# ASCENSION: Autoencoder-Based Latent Space Class Expansion for Time Series Data Augmentation

## Abstract

Achieving effective data augmentation (DA) in time series (TS) classification is challenging due to the complex nature of temporal data. While state-of-the-art generative models for DA, based on generative adversarial networks, diffusion models and variational autoencoders (VAEs), demonstrate potential, they often fail to yield consistent performance gains across diverse domains (e.g., ECG, power, vibration). To overcome this, we propose **ASCENSION** (**A**utoencoder-based latent **S**pace **C**lass **ExpaNSION**), a novel generative framework that leverages the probabilistic nature of a VAE latent space together with a contrastive loss to promote intra-class compactness and inter-class separability. Its key innovation is an $\alpha$-scaling mechanism that progressively expands per-class posterior covariances while preserving class identity, populating under-represented neighbourhoods beyond the training distribution. We evaluate ASCENSION on 102 univariate datasets from the UCR benchmark using two established deep TS classifiers and a recent TS foundation model, comparing it against eight state-of-the-art DA methods. Empirical results demonstrate that ASCENSION increases average classification accuracy by approximately 2%, while the strongest baseline method results in a $-1.7\%$ decrease. Notably, ASCENSION delivers non-negative performance gains on 73.5% of datasets (averaged over the three classifiers), compared to 50.0% for the second best-performing baseline. An ablation study further highlights the significant impact of the $\alpha$-scaling mechanism on these gains. These findings position ASCENSION as the only DA method in our benchmark that delivers consistent positive gains across all three classifier families on the 102-dataset UCR archive, without requiring prior knowledge of method suitability.

## 1 Introduction

Time series (TS) classification (TSC) is challenging due to temporal dependencies, non-stationarity, or scarce labelled data, often limited by acquisition costs and privacy. Data augmentation (DA) mitigates these issues by enriching the training set with synthetic samples. The practical challenge for DA in TSC is not that no method helps on some datasets, many do, but that no single method helps *reliably* across the heterogeneous domains encountered in practice (ECG, accelerometer, image-derived, spectral, motion, power). Empirical studies (Iwana & Uchida, 2021; Iglesias et al., 2023) document this inconsistency: the technique that wins on one dataset can degrade performance on the next. Practitioners, therefore, face real overhead in testing several DA methods on every new dataset to find one that does not hurt.

Existing in-distribution generative DA methods for TSC learn to model the training distribution faithfully but, by design, rarely produce samples that extend beyond its empirical support, leaving the decision boundary under-supplied near its hardest regions, precisely where augmentation could help the most.

We posit that progressively exploring under-represented regions of the latent space by expanding the initial data distribution can enhance DA effectiveness, particularly under train/test discrepancy. To this end, we propose ASCENSION, a DA framework that leverages a variational autoencoder (VAE) architecture and introduces a tunable $\alpha$-scaling mechanism to gradually explore these under-represented regions while

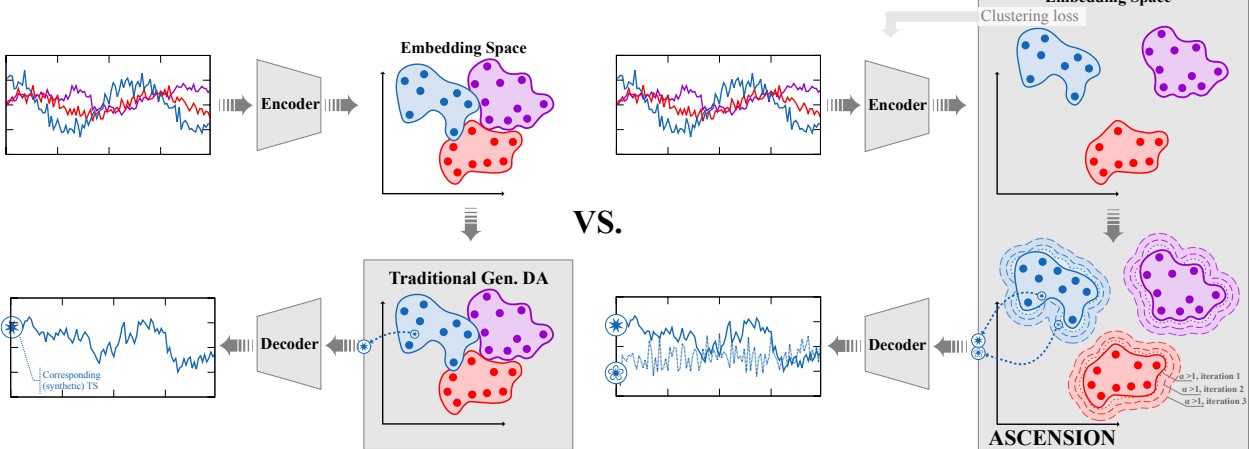

Figure 1: Comparison of latent space dynamics between state-of-the-art generative DA methods and AS-CENSION. State-of-the-art methods generate samples confined to the learned distribution, limiting diversity (left). ASCENSION (right) introduces a novel $\alpha$-scaling mechanism, controlled by a parameter $\alpha$, which progressively expands class regions, promoting intra-class compactness, inter-class separation, and reduced overlap during sampling.

maintaining intra-class compactness and inter-class separation. Figure 1 illustrates the core concept. This strategy yields more diverse synthetic data and improved classification performance, as shown in Section 4.1.2.

The key contributions of this paper are:

- **Novel VAE-based DA method.** A progressive, controllable boundary-expansion strategy via an $\alpha$-scaling mechanism, with ablations mapping the dataset conditions where it is most beneficial.

- **Comprehensive benchmark & performance gains.** ASCENSION is evaluated across 102 UCR datasets against one traditional (FAA) and eight generative baselines (Diffusion-TS, ImagenTime, KoVAE, LatentAugment, TTS-GAN, Time-DDPM, VaDE, MODALS), with consistent gains on a majority of the datasets.

- **Data-driven insights.** We investigate how ASCENSION's components and TS features influence DA effectiveness, showing that $\alpha$-scaling is the single most impactful component.

## 2 Problem Setup

This section formalises the TS DA setting and surveys the landscape of existing generative DA methods. Section 2.1 fixes notation; the remainder positions ASCENSION in the broader literature.

### 2.1 Time-series data augmentation

DA aims to expand the training set with synthetic samples to increase the generalisation ability of the trained model (Shorten & Khoshgoftaar, 2019), primarily by reducing overfitting, which we measure operationally in our setting as performance on the test set. Formally, let $\mathcal{D} = \{(x_i, y_i)\}_{i=1}^N$ be a labelled TS dataset with $x_i \in \mathbb{R}^T$ a univariate series of length $T$ and $y_i \in \{1, \ldots, Y\}$ a class label. A DA method produces an auxiliary set $\mathcal{D}' = \{(x'_j, y'_j)\}_{j=1}^M$. The objective is for a classifier trained on $\mathcal{D} \cup \mathcal{D}'$ to reach higher test-set accuracy than one trained on $\mathcal{D}$ alone (or at least not lower).

One of the main challenges for DA in TSC is not that no method helps on some datasets, but that no single method helps reliably across the heterogeneous domains encountered in practice (ECG, accelerometer, image-derived, spectral, motion, power). Empirical studies (Iwana & Uchida, 2021; Iglesias et al., 2023) have

Figure 2: Overview of state-of-the-art DA methods for TS (traditional vs. generative). **MODALS\*\***: Although the code was released 6 years ago, it is now non-functional; the authors confirmed it cannot be restored without major re-coding. We therefore compare ASCENSION and MODALS on the HAR dataset used in Cheung & Yeung (2020).

documented this inconsistency: DA methods that win on one dataset (resp. domain) degrade accuracy on the next. As a result, practitioners face the real overhead of testing several DA methods on each new dataset to find one that does not hurt, a cost that scales with the number of methods available and grows as the literature expands. ASCENSION is designed to reduce that cost. Therefore, our evaluation jointly targets: (i) the average gain, as the method should improve mean test-set accuracy across a large benchmark, (ii) the gain consistency, so that it can be adopted without a per-dataset method-selection loop.

We report both metrics in Section 4.1.2.

## 2.2 Related work

TS DA methods can be classified as traditional or generative according to (Iglesias et al., 2023; Iwana & Uchida, 2021) and summarised in Figure 2.

**Traditional DA methods** Traditional DA methods, such as window slicing, jittering, and scaling, are primarily adapted from computer vision and rely on transformation strategies like cropping, rotation, scaling, drifting, and so forth. However, the complex nature of TS data often renders these methods sub-optimal, as they can disrupt the semantic integrity of the original data. For instance, while a slightly flipped image of a cat remains recognisable, reversing the time axis of an electrocardiogram sequence can render it meaningless. In response to these challenges, more advanced DA techniques were developed to automate the sequence of transformations to be performed. The first method, named **AutoAugment (AA)** (Cubuk et al., 2019), uses reinforcement learning to explore transformation pipelines/policies. The second, named **Fast AutoAugment (FAA)** (Lim et al., 2019), uses density matching as a faster search strategy, eliminating the need for back-propagation. Subsequent methods, such as **RandAugment** (Cubuk et al., 2020), **Deep AutoAugment** (Zheng et al., 2022), and **Trivial Augment** (Müller & Hutter, 2021), were introduced to further simplify and refine the augmentation search strategy. RandAugment streamlines the augmentation process by removing the exhaustive search phase, instead applying a fixed number of random transformations with adjustable magnitudes. Deep AutoAugment incorporates a deep RL model that dynamically combines transformation policies based on the specific characteristics of the dataset. Trivial Augment introduces an even simpler approach by applying a minimal set of random transformations, emphasising ease of use and computational efficiency. Despite these advancements, the aforementioned methods rely on predefined transformations, which is suboptimal for preserving intra-class consistency and the semantic characteristics of the original TS data, thereby limiting the effectiveness of DA.

**Generative DA methods** Generative DA models such as Generative Adversarial Networks (GANs) (Goodfellow et al., 2020), diffusion models (Yang et al., 2023a), and VAEs (Kingma & Welling, 2014) represent powerful techniques capable of learning a probabilistic representation of data distributions. These models can generate TS data that retain the temporal dependencies, semantic consistency, and class-specific characteristics of the original datasets (Fu et al., 2020). For example, using a representation layer, as introduced by Liu et al. (2023), provides an abstraction that is crucial when dealing with TS data. **TimeGAN** (Zhang et al., 2022), designed specifically for TS, shows significant improvements in generating high-quality synthetic sequences and augmenting low-quality datasets. Likewise, **TS-GAN** (Yang et al., 2023b) develops an LSTM-based GAN architecture with a sequential-squeeze-and-excitation to better capture time-dependence between the current and past moments in each dimension. TS-GAN is proposed to generate augmented sensor-based health data to improve Deep Learning classification models and is evaluated on three health TS datasets. **GT-GAN** (Jeon et al., 2022) introduces an invertible GAN framework that combines controlled differential equations and continuous-time flow processes, showing promising results. **TTS-GAN** (Li et al., 2022a) and its conditional variant **TTS-cGAN** (Li et al., 2022b) adapt the traditional GAN architecture using a transformer-encoder architecture that can deal with long-range dependencies in time sequences. They show strong performance in generating realistic data across three datasets: a simulated dataset, a human activity recognition dataset, and an ECG dataset. **LatentAugment** (Tronchin et al., 2023) guides the latent space learned by any GAN on the original data, applies noise-based perturbations to the latent points, and decodes them to generate new, semantically consistent samples. More recently, Seon et al. (2025) proposed **LISGAN**, a GAN-based architecture for TS augmentation under class imbalance: they augment the loss with a mutual information term and employ spectral normalisation. LISGAN yields high-quality synthetic data and significantly improves classification performance on industrial IoT datasets. A recent extension, **TTS-WGAN-GP** (Mousavi et al., 2025), leverages a transformer architecture within a Wasserstein GAN with gradient penalty framework to improve DA quality for structural health monitoring TS. However, training GANs is notoriously unstable and highly sensitive to hyperparameter settings. Without careful tuning, they often suffer from mode collapse, which reduces sample diversity and leads to unrealistic data distributions (Lei et al., 2019).

Diffusion models generate data by iteratively denoising noise toward the target distribution, rather than via the adversarial training used by GANs. They have achieved high-fidelity image generation (e.g., DALL·E 2, Imagen, Flux). Since 2023, several diffusion-based DA methods for TS have appeared, including **SE-DDPM** (Liu et al., 2024) for imbalanced TSC, **DiffRUL** (Wang et al., 2024) for remaining useful life prediction, **D3A-TS** (Solis-Martin et al., 2023) for meta-attribute conditioning, **Time-DDPM** (Dai et al., 2023) combining diffusion with CNN-LSTM, **Diffusion-TS** (Yuan & Qiao, 2024) with seasonal-trend decomposition, and **ImagenTime** (Naiman et al., 2024a) transforming sequences into images. **DiffAT** (Yu et al., 2025) introduces a flexible method that uses soft prompts distilled from traditional DA to guide conditional diffusion for TS forecasting. **Diff-TI** (Zhang et al., 2025) proposes a hybrid diffusion-transformer architecture where diffusion generates the initial timestep (capturing the base distribution), followed by iterative transformer prediction for the next timesteps. Despite stable outputs, diffusion models still face challenges with long-range dependencies, error accumulation, and slow inference (Feng et al., 2024), which can limit their practical applications.

VAEs allow for explicit control over the diversity of generated samples through manipulation of the latent space, as evidenced by Cheung & Yeung (2020). Additionally, VAEs are less prone to collapse compared to GANs and are less computationally expensive than both GANs and diffusion models (Thanh-Tung & Tran, 2020). The first VAE-based generative DA model relying on clustering, named **VaDE**, was introduced by Jiang et al. (2017). VaDE integrates a Gaussian Mixture Model (GMM) as the prior distribution, jointly trained with the encoder on unlabelled data, enabling realistic sample generation for any specified cluster without supervised labels. **GMVAE** (Dilokthanakul et al., 2017) similarly employs a GMM-structured latent space to capture multi-modal data distributions. In both cases, the mixture is an unsupervised inductive bias: it is part of the generative model and appears in the evidence lower bound (ELBO), and neither method provides a mechanism to control the extent of generation beyond the learned distribution. **VAE-STS** (Goubeaud et al., 2021) proposes a simple VAE architecture for data generation. **KoVAE** (Naiman et al., 2024b) uses linear Koopman dynamics as a prior, enabling interpretable and physics-constrained modelling of both regular and irregular TS. **MODALS** was introduced by Cheung & Yeung (2020) and

was the first study to explore class boundary expansion during synthetic data generation, though it lacks a mechanism to control the extent of this expansion. Recently, Dang et al. (2024) introduced **VAE-LSTM** to augment an inertial sensor dataset with the goal of enhancing classification performance. **Seq-MVAE** (Bogojeski et al., 2025) employs a VAE with multimodal heterogeneous TS for industrial aging processes, enabling DA that preserves cross-modal temporal dependencies.

However, none of these approaches explore a progressive, tunable expansion of class representations in the latent space, as proposed in ASCENSION.

## 3 Proposed Method: ASCENSION

In this section, we propose ASCENSION, a new DA framework for TSC. We first identify why in-distribution generative augmentation underperforms across diverse datasets (Section 3.1). We then describe the three ingredients that allow ASCENSION to address this failure mode: a VAE backbone (Section 3.2), a contrastive clustering constraint that keeps classes compact and separated (Section 3.2), and an $\alpha$-scaling mechanism that expands each class's latent density into under-represented neighbourhoods (Section 3.3).

### 3.1 Why in-distribution generative augmentation underperforms

Existing generative DA methods for TSC learn to model the training distribution faithfully, but by design, rarely produce samples that extend beyond its empirical support. When the test distribution drifts away from training (through sample scarcity, distributional shift, or class regions that the encoder under-represents), augmentation strictly within the empirical support cannot supply the classifier with examples near the parts of the decision boundary that matter most. ASCENSION follows directly from this observation. Rather than mimicking the training posterior, we propose expanding it in a controlled, class-aware manner so that augmented samples are more likely to populate the under-represented latent neighbourhoods of each class. However, expanding the empirical support is geometrically precarious: naive scaling pushes samples across class boundaries, causing label-flipping. Safe expansion therefore requires an explicit geometric account of these boundaries, which we formalise through a variance decomposition in Section 3.3.

### 3.2 VAE backbone and training objective

ASCENSION adopts a VAE architecture for its ability to flexibly capture the data distribution in a learned latent space. Given a labelled dataset $\mathcal{D} = \{(x_i, y_i)\}_{i=1}^{N}$ with $y_i \in \{1, \ldots, Y\}$, we denote by $\mathcal{Z} \subseteq \mathbb{R}^{d_z}$ the $d_z$-dimensional latent space. The encoder defines an approximate posterior $q_\phi(z \mid x) = \mathcal{N}(z; \mu_\phi(x), \Sigma_\phi(x))$ over $\mathcal{Z}$, with diagonal covariance $\Sigma_\phi(x) = \mathrm{diag}(\sigma_\phi(x)^2)$ (where $\sigma_\phi(x)$ is the encoder's standard-deviation vector), and the decoder $p_\theta(x \mid z)$ defines a likelihood under the prior $p(z) = \mathcal{N}(0, I_{d_z})$. We deliberately avoid class conditioning (cVAE), which can amplify posterior collapse when $x$ is highly correlated with $y$ (Dang et al., 2023); an empirical comparison with cVAE baselines is reported in Appendix D. To encourage class-discriminative geometry, we additionally attach an auxiliary classifier head $h_\psi : \mathcal{Z} \to \mathbb{R}^Y$ on the latent code, trained jointly with the VAE.

**Training objective.** ASCENSION is trained by minimising a weighted sum of four terms,

$$\mathcal{L}_{\mathrm{VAE}} = \underbrace{\gamma_1 \, \mathcal{L}_{\mathrm{recon}} + \gamma_2 \, \mathcal{L}_{\mathrm{KL}}}_{\text{negative ELBO}} + \underbrace{\gamma_3 \, \mathcal{L}_{\mathrm{cluster}}}_{\text{contrastive clustering}} + \underbrace{\gamma_4 \, \mathcal{L}_{\mathrm{class}}}_{\text{auxiliary classifier}}, \tag{1}$$

with non-negative weights $\gamma_i$. The first two terms are the standard ELBO components: a reconstruction loss $\mathcal{L}_{\mathrm{recon}} = \|x - \mu_\theta(z)\|_2^2$, where $\mu_\theta(z)$ is the decoder mean, and a Kullback-Leibler (KL) divergence $\mathcal{L}_{\mathrm{KL}} = D_{\mathrm{KL}}(q_\phi(z \mid x) \| p(z))$ that regularises the posterior toward the prior. The third term is the margin-based contrastive loss of Hadsell et al. (2006), applied here to the per-sample posterior means $z_i = \mu_\phi(x_i)$,

$$\mathcal{L}_{\mathrm{cluster}} = \frac{1}{N^2} \sum_{i,j} \mathbf{1}_{y_i = y_j} \, d_{ij}^2 + \mathbf{1}_{y_i \neq y_j} \, \max(0, m - d_{ij})^2, \quad d_{ij} = \|z_i - z_j\|_2, \tag{2}$$

which pulls same-class means together and pushes different-class means apart by at least the margin $m > 0$. The fourth term is the cross-entropy between the auxiliary classifier $h_\psi$ and the labels,

$$\mathcal{L}_{\text{class}} = -\frac{1}{N} \sum_{i=1}^{N} \log \text{softmax}\big(h_\psi(z_i)\big)_{y_i}, \quad z_i \sim q_\phi(z \mid x_i), \tag{3}$$

The contrastive loss shapes the inter-class geometry of posterior means, while the classifier head extends this discriminative pressure to the per-sample posterior; together they produce the latent geometry that Section 3.3 exploits.

### 3.3 Latent class expansion

The goal of ASCENSION's augmentation step is to expand each class's latent distribution into under-represented neighbourhoods while preserving class identity. We define (i) an $\alpha$-scaled class density that controls within-class spread, and (ii) a Bayes region that the rejection step uses to enforce class consistency.

**Definition 1** ($\alpha$-scaled class density). For class $y \in \{1, \ldots, Y\}$, let $\mathcal{I}_y = \{i : y_i = y\}$ index the training samples with label $y$. The $\alpha$-*scaled class density* is the uniform mixture of encoder posteriors with covariance scaled by $\alpha \geq 1$:

$$d_y^\alpha(z) = \frac{1}{|\mathcal{I}_y|} \sum_{i \in \mathcal{I}_y} \mathcal{N}\big(z; \mu_\phi(x_i), \alpha \Sigma_\phi(x_i)\big). \tag{4}$$

**Definition 2** (Bayes region). For class $y$ and $\alpha \geq 1$,

$$\mathcal{R}_y^\alpha = \big\{ z \in \mathcal{Z} : y = \arg\max_{y' \in \{1, \ldots, Y\}} d_{y'}^\alpha(z) \big\}. \tag{5}$$

$\mathcal{R}_y^\alpha$ is the latent region where $d_y^\alpha$ dominates the other class mixture densities, equivalently the maximum-a-posteriori region under a uniform class prior.

**Why both ingredients are required.** The $\alpha$-scaling and contrastive losses are not independent levers: they interact through the geometry of the Bayes region. Proposition 1 makes this precise by decomposing the variance of the sampling density.

**Proposition 1** (Variance decomposition of $d_y^\alpha$). *Let* $\bar{\mu}_y := \frac{1}{|\mathcal{I}_y|} \sum_{i \in \mathcal{I}_y} \mu_\phi(x_i)$ *denote the empirical mean of the per-sample posterior means in class* $y$. *For any* $\alpha \geq 0$,

$$\text{Cov}_{Z \sim d_y^\alpha}[Z] = \alpha \underbrace{\frac{1}{|\mathcal{I}_y|} \sum_{i \in \mathcal{I}_y} \Sigma_\phi(x_i)}_{\bar{\Sigma}_w(y) \ (within\text{-}class \ spread)} + \underbrace{\frac{1}{|\mathcal{I}_y|} \sum_{i \in \mathcal{I}_y} \big(\mu_\phi(x_i) - \bar{\mu}_y\big)\big(\mu_\phi(x_i) - \bar{\mu}_y\big)^\top}_{\Sigma_b(y) \ (between\text{-}class \ spread)}. \tag{6}$$

*The proof is deferred to Appendix C.*

Equation 6 attributes the spread of $d_y^\alpha$ to two distinct sources: encoder noise scaled by $\alpha$ (first term) and the within-class dispersion of posterior means (second term, $\alpha$-independent). The contrastive loss $\mathcal{L}_{\text{cluster}}$ shapes both: its same-class component drives $\Sigma_b(y)$ small (each class collapses to a tight cluster of posterior means). Its different-class component pushes centroids of distinct classes apart by an inter-class margin $m$ (not appearing in Equation 6 but still constraining the layout around $\mathcal{R}_y^\alpha$).

Without $\mathcal{L}_{\text{cluster}}$, $m \to 0$ and any $\alpha$-inflated sample crosses into a neighbouring Bayes region. Therefore, the rejection step in Algorithm 1 discards almost all candidates and the $\alpha$-mechanism degenerates to noise. Figure 3 visualises the resulting acceptance landscape, and the ablation in Section 4.2 (Table 4) confirms the prediction empirically: removing $\mathcal{L}_{\text{cluster}}$ collapses the mean augmentation gain to near zero across the 102 UCR datasets.

Figure 3 shows the ratio $m/\sigma_w$ on its $y$-axis, where $m$ is the inter-class centroid distance enforced by $\mathcal{L}_{\text{cluster}}$ and $\sigma_w = \sqrt{\lambda_{\max}(\bar{\Sigma}_w(y))}$ is the within-class radius along the most-spread principal direction of $\bar{\Sigma}_w(y)$ ($\lambda_{\max}$ denotes the largest eigenvalue). We plot this ratio rather than $m$ alone because the absolute scale of the latent space depends on encoder hyperparameters. In practice $\sigma_w \approx 1$ (the $\mathcal{N}(0, I)$ prior pulls the posterior toward unit variance).

**Augmentation procedure.** With the geometry in place, augmentation reduces to sampling from $d_y^\alpha$ and rejecting candidates that fall outside $\mathcal{R}_y^\alpha$. At each iteration, the VAE is first trained using $\mathcal{L}_{\text{VAE}}$ (Eq. 1) on the current dataset. Candidate latent points are then drawn from $d_y^\alpha$ and accepted iff they lie in the source class's $\mathcal{R}_y^\alpha$ (the maximum a posteriori rule under a uniform class prior). Accepted candidates are decoded through $p_\theta(x \mid z)$ and folded into the training set; the loop iterates by retraining the model from scratch on the augmented dataset. The full procedure is given in Algorithm 1.

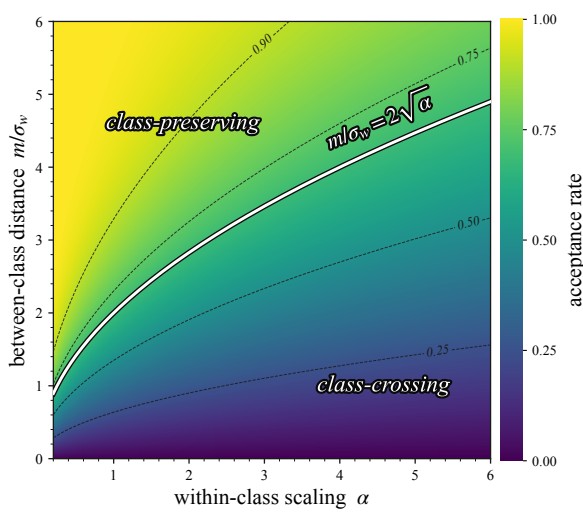

Figure 3: Acceptance landscape under Proposition 1. The colour shows $\mathbb{P}_{Z \sim d_y^\alpha}[Z \in \mathcal{R}_y^\alpha]$ on the $(\alpha, m/\sigma_w)$ plane. Bold curve $m/\sigma_w = 2\sqrt{\alpha}$ traces where the $\alpha$-scaled within-class spread $\sqrt{\alpha}\,\sigma_w$ equals the half-distance $m/2$ to the nearest neighbour.

---

**Algorithm 1** ASCENSION augmentation loop

---

**Require:** Original TS data $\mathbf{X} = \{x_1, \ldots, x_n\}$ with labels $\mathbf{Y} = \{y_1, \ldots, y_n\}$
**Ensure:** Augmented datasets $\mathbf{X}_{\text{aug}}, \mathbf{Y}_{\text{aug}}$
 1: $\mathbf{X}_{\text{aug}} \leftarrow \mathbf{X}, \ \mathbf{Y}_{\text{aug}} \leftarrow \mathbf{Y}$
 2: **while** augmentation desired **do**
 3:     $\theta^*, \phi^* \leftarrow \arg\min_{\theta, \phi} \mathcal{L}_{\text{VAE}}$ using $(\mathbf{X}_{\text{aug}}, \mathbf{Y}_{\text{aug}})$
 4:     **for all** classes $y$ **do**
 5:         Sample $\mathbf{Z}_{\text{new}}^y \sim d_y^\alpha$                                ▷ Definition 1
 6:         $\mathbf{Z}_{\text{accept}}^y \leftarrow \{ z' \in \mathbf{Z}_{\text{new}}^y : z' \in \mathcal{R}_y^\alpha \}$              ▷ Definition 2
 7:         Decode $x' = \mu_{\theta^*}(z')$ for all $z' \in \mathbf{Z}_{\text{accept}}^y$
 8:     **end for**
 9:     Append decoded $(x', y)$ pairs to $(\mathbf{X}_{\text{aug}}, \mathbf{Y}_{\text{aug}})$
10: **end while**

---

# 4 Experiments

In this section, we evaluate ASCENSION against leading state-of-the-art DA methods to validate our central hypothesis that progressively exploring under-represented latent regions and expanding the initial data distribution improves DA, particularly under distribution discrepancies.

## 4.1 Experimental setup

We use the UCR Archive of 128 univariate TSC datasets (see Table A), retaining 102 after excluding the 26 smallest to ensure sufficient training data.

Following (Ismail Fawaz et al., 2019), which identifies ResNet-50 and FCN (Fully Convolutional Network) as top UCR performers, we use these architectures as in (Koonce, 2021; Scabini & Bruno, 2023). We also evaluate Moment (Goswami et al., 2024), a transformer-based multi-task foundation model pretrained on the UCR training splits (excluding test splits). We apply ASCENSION to augment the training data and

assess whether augmentation still helps an encoder that has already seen this data. As the test splits are held out from Moment's pretraining, any test-set improvement reflects a performance benefit.

We compare ASCENSION to nine related DA methods: one traditional (FAA) and eight generative (VaDE, TTS-GAN, KoVAE, Time-DDPM, LA, ImagenTime, Diffusion-TS, MODALS; see Section 2.2 and Figure 2 for details). FAA provides a classical baseline, while KoVAE, VaDE, and MODALS share architectural traits with ASCENSION. Diffusion-TS, ImagenTime, LA, Time-DDPM, and TTS-GAN are recent generative baselines. As MODALS' public code is defunct and unmaintained, we benchmark it on the HAR dataset from (Cheung & Yeung, 2020) instead of UCR.

Table 1 reports ASCENSION at its best hyperparameter configuration per dataset; baselines use their authors' recommended hyperparameters, and per-epoch model selection follows the best-epoch-on-test convention standard in the UCR literature (Ismail Fawaz et al., 2019). A single fixed configuration ($\alpha = 1.5$, 1 iteration) for ASCENSION is reported in Appendix E, where ASCENSION remains the only method with a positive total improvement on every classifier.

### 4.1.1 Performance evaluation metrics

We rely on the UCR archive protocol (Dau et al., 2019) and use classification accuracy as the primary metric, quantifying each DA method's gain as the change in accuracy before and after augmentation. Table 1 reports the number of datasets with improved, unchanged, or worsened accuracy, along with mean post-augmentation accuracy; full results appear in Appendix J.

### 4.1.2 Performance comparison analysis

Table 1 shows that FAA yields high mean gains on its winning datasets ($7.8\%, 7.1\%, 7.8\%$ for ResNet, FCN, Moment) but is highly inconsistent, improving only 19/102, 14/102, and 50/102 datasets, with severe degradations ($-9.2\%, -15.0\%, -11.8\%$) on the rest. KoVAE is the most balanced of the competitors, improving 38 (ResNet), 41 (FCN), 27 (Moment) datasets with the smallest mean drops on ResNet/FCN ($-2.1\%, -2.1\%$), but its net gains are below zero on every classifier ($-0.4\%, -0.1\%, -7.4\%$). LA performs comparably to KoVAE on FCN (44/39 improved/worsened, net 0.0%) but struggles on ResNet (14/75, $-2.4\%$). VaDE and TTS-GAN show similar patterns: moderate consistency on a third of datasets but net-negative on every classifier. Time-DDPM has the highest mean gains on its winning datasets (16.3% ResNet, 16.9% FCN) but severe drops ($-22.3\%, -23.0\%$) on the majority, yielding the worst total on ResNet ($-7.9\%$) and a near-worst on FCN ($-6.3\%$); on Moment, large-scale pretraining mitigates these failures ($-2.9\%$). In contrast, ASCENSION combines competitive mean gains (4.1% ResNet, 4.0% FCN, 3.8% Moment) with by far the highest improved-dataset count (59, 56, 72) and the lowest degradation count (27, 34, 20). It is the only method with a positive total across all three classifiers ($+1.9\%$ ResNet, $+1.4\%$ FCN, $+2.4\%$ Moment), and the only one whose mean drop on losing datasets stays below $-3\%$ on every classifier. The $+2.4\%$ Moment gain remains in the same single-digit-percent range as ResNet/FCN, indicating that ASCENSION still adds value on top of an encoder pretrained on these UCR training splits, rather than being competed away by that prior exposure.

In Table 2, we compare ASCENSION to MODALS on the HAR dataset, as UCR benchmarking is not feasible for MODALS (see Section 4.1). Using MODALS improves baseline accuracy by 3.23%, while ASCENSION achieves a 4.78% improvement over the baseline.

### 4.1.3 Embedded classifier performance

Table 1 reports the *default* ASCENSION configuration: an external classifier (ResNet, FCN, or Moment) is trained on the *decoded* synthetic samples. ASCENSION's training objective also includes an auxiliary classifier head $\mathcal{L}_{\text{class}}$ on the latent code (Section 3.2), which enables two alternative configurations that bypass the decoder. $ASCENSION_{Emb}$ uses only this internal classifier head, with no external classifier. $ASCENSION_c$, with $c \in \{\text{ResNet}, \text{FCN}\}$, attaches an external classifier directly to the latent space (after encoder and reparameterisation) rather than to the decoder output. Table 3 reports both alternative configurations; the ablation studies in Section 4.2 use ASCENSION$_{\text{ResNet-Emb}}$ and ASCENSION$_{\text{FCN-Emb}}$. For the embedded

Table 1: For each DA method, we show the number of datasets (Nb$_{\text{data}}$), *out of 102*, with improved, unchanged, or worsened accuracy, plus the mean accuracy change $\overline{\Delta\text{Acc}}$ = augmentation_accuracy − base_accuracy averaged over each subset. Arrows ($\uparrow, \downarrow$) mark direction of improvement; **bold** and underline indicate **best** and second best.

| | DA method | (Venue, Year) | Improved | | Unchanged | | Worsened | | Total | |
|---|---|---|---|---|---|---|---|---|---|---|
| | | | $\uparrow$Nb$_{\text{data}}$ | $\uparrow\overline{\Delta\text{Acc}}$ | Nb$_{\text{data}}$ | $\overline{\Delta\text{Acc}}$ | $\downarrow$Nb$_{\text{data}}$ | $\downarrow\overline{\Delta\text{Acc}}$ | Nb$_{\text{data}}$ | $\overline{\Delta\text{Acc}}$ |
| ResNet | VaDE | (IJCAI, 2016) | 44 | 2.8% | 9 | 0% | 49 | -7.1% | 102 | $-2.2\%$ |
| | FAA | (NeurIPS, 2019) | 19 | 7.8% | 8 | 0% | 75 | -9.2% | 102 | $-5.3\%$ |
| | TTS-GAN | (AIME, 2022) | 26 | 2.2% | 11 | 0% | 65 | -8.3% | 102 | $-4.7\%$ |
| | KoVAE | (ICLR, 2024) | 38 | 1.5% | 19 | 0% | 45 | -2.1% | 102 | $\underline{-0.4\%}$ |
| | Time-DDPM | (SENS. J., 2023) | 38 | **16.3%** | 0 | 0% | 64 | -22.3% | 102 | $-7.9\%$ |
| | LA | (ArXiv, 2023) | 14 | 3.3% | 13 | 0% | 75 | -3.9% | 102 | $-2.4\%$ |
| | ImagenTime | (NeurIPS, 2024) | 15 | 1.3% | 13 | 0% | 74 | -6.1% | 102 | $-4.2\%$ |
| | Diffusion-TS | (ICLR, 2024) | 21 | 1.3% | 5 | 0% | 76 | -9.3% | 102 | $-6.7\%$ |
| | **ASCENSION** | | **59** | 4.1% | 16 | 0% | **27** | **-1.9%** | 102 | $+\mathbf{1.9\%}$ |
| FCN | VaDE | (IJCAI, 2016) | 42 | 3.4% | 15 | 0% | 45 | -7.0% | 102 | $-1.7\%$ |
| | FAA | (NeurIPS, 2019) | 14 | 7.1% | 3 | 0% | 85 | -15.0% | 102 | $-11.5\%$ |
| | TTS-GAN | (AIME, 2022) | 36 | 2.9% | 17 | 0% | 49 | -8.2% | 102 | $-2.9\%$ |
| | KoVAE | (ICLR, 2024) | 41 | 2.1% | 15 | 0% | 46 | **-2.1%** | 102 | $-0.1\%$ |
| | Time-DDPM | (SENS. J., 2023) | 42 | **16.9%** | 1 | 0% | 59 | -23.0% | 102 | $-6.3\%$ |
| | LA | (ArXiv, 2023) | 44 | 2.3% | 19 | 0% | 39 | -2.6% | 102 | $\underline{0.0\%}$ |
| | ImagenTime | (NeurIPS, 2024) | 38 | 2.7% | 11 | 0% | 53 | -2.9% | 102 | $-0.5\%$ |
| | Diffusion-TS | (ICLR, 2024) | 36 | 3.3% | 7 | 0% | 59 | -13.5% | 102 | $-6.6\%$ |
| | **ASCENSION** | | **56** | 4.0% | 12 | 0% | **34** | -2.5% | 102 | $+\mathbf{1.4\%}$ |
| Moment | VaDE | (IJCAI, 2016) | 35 | 2.9% | 8 | 0% | 59 | -5.6% | 102 | $-2.2\%$ |
| | FAA | (NeurIPS, 2019) | 50 | **7.8%** | 6 | 0% | 46 | -11.8% | 102 | $-1.5\%$ |
| | TTS-GAN | (AIME, 2022) | 41 | 3.4% | 7 | 0% | 54 | -5.2% | 102 | $\underline{-1.4\%}$ |
| | KoVAE | (ICLR, 2024) | 27 | 3.7% | 5 | 0% | 70 | -12.2% | 102 | $-7.4\%$ |
| | Time-DDPM | (SENS. J., 2023) | 35 | 3.8% | 5 | 0% | 62 | -7.0% | 102 | $-2.9\%$ |
| | LA | (ArXiv, 2023) | 31 | 2.6% | 6 | 0% | 65 | -5.5% | 102 | $-2.7\%$ |
| | ImagenTime | (NeurIPS, 2024) | 25 | 2.1% | 7 | 0% | 70 | -6.3% | 102 | $-3.8\%$ |
| | Diffusion-TS | (ICLR, 2024) | 28 | 3.2% | 5 | 0% | 69 | -8.8% | 102 | $-5.1\%$ |
| | **ASCENSION** | | **72** | 3.8% | 10 | 0% | **20** | **-1.6%** | 102 | $+\mathbf{2.4\%}$ |

Table 2: ASCENSION and MODALS performance on HAR dataset from (Cheung & Yeung, 2020).

| Method | **ASCENSION** | **MODALS** | **No Aug** |
|---|---|---|---|
| Accuracy (%) | **93.42** | 91.87 | 88.64 |

variants, augmentation gain is defined as the accuracy difference between the best of ASCENSION$_{\text{Emb}}$ and ASCENSION$_c$ and the no-DA baseline:

$$\text{Acc}_{\text{Emb},c} = \max_{k \in \{\text{Emb},c\}} \text{Acc}_{\text{ASCENSION}_k} - \text{Acc}_{\text{base}}. \tag{7}$$

Table 3: Number of datasets with improved, unchanged, or worsened accuracy per DA for each embedded classifier; mean accuracy change ($\uparrow\overline{\Delta\text{Acc}}$) per category. Arrows ($\uparrow, \downarrow$) show value preference.

| Embedded Classifier | Improved | | Unchanged | | Worsened | | Total | |
|---|---|---|---|---|---|---|---|---|
| | $\uparrow$Nb$_{\text{data}}$ | $\uparrow\overline{\Delta\text{Acc}}$ | Nb$_{\text{data}}$ | $\overline{\Delta\text{Acc}}$ | $\downarrow$Nb$_{\text{data}}$ | $\downarrow\overline{\Delta\text{Acc}}$ | Nb$_{\text{data}}$ | $\overline{\Delta\text{Acc}}$ |
| ASCENSION$_{\text{Emb}}$ | 66 | 1.9% | 12 | 0% | 24 | **-1.6%** | 102 | $+0.8\%$ |
| ASCENSION$_{\text{ResNet-Emb}}$ | **76** | **3.7%** | 12 | 0% | 14 | -5.7% | 102 | $+\mathbf{2.1\%}$ |
| ASCENSION$_{\text{FCN-Emb}}$ | 60 | 2.9% | 28 | 0% | 14 | -1.7% | 102 | $+1.2\%$ |

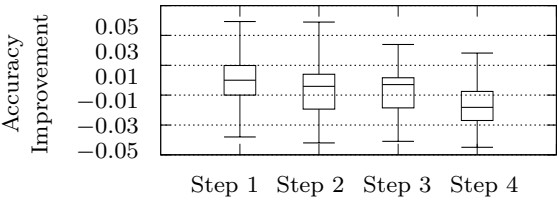

(a) Step-wise accuracy degradation from clustering loss removal.

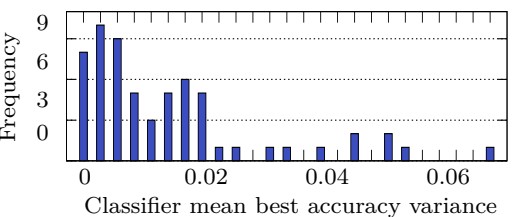

(b) Averaged accuracy variance per dataset over $\gamma_i$. Smaller values imply low impact.

Figure 4: Results on clustering loss impact across augmentation steps and loss weight tuning.

Table 3 shows ASCENSION$_{\text{ResNet-Emb}}$ yields the highest gain (3.7% on 76 datasets) with the largest drop ($-5.7\%$ on 14). ASCENSION$_{\text{Emb}}$ is steadier (1.9% gain, $-1.6\%$ drop), while ASCENSION$_{\text{FCN-Emb}}$ offers moderate gains (2.9%) with balanced trade-offs. Overall, ResNet yields the largest gains but is less stable; FCN and the default classifier are more consistent.

### 4.2 Ablation 1: Hyperparameter analysis & tuning

We perform a unified ablation on ASCENSION$_{\text{ResNet-Emb}}$ to assess key components and hyperparameters: (i) We sweep $\alpha \in \{1, \ldots, 5\}$ (with $\alpha = 1$ disabling scaling) and augmentation iterations $\{1, \ldots, 9\}$ to study the $\alpha$-scaling mechanism and performance gains evolution; (ii) We test the effect of removing the clustering loss by setting $\gamma_3 = 0$; (iii) We grid-search $(\gamma_1, \gamma_2, \gamma_3, \gamma_4) \in G^4$, with $G = \{1, 3, 5, 8, 10\}$, to find optimal loss-weighting. All experiments use the protocol of Section 4.1.3, isolating the impact of each factor on classification accuracy.

**$\alpha$-scaling mechanism ablation** Table 4 reports the accuracy improvements (Q1, median, Q3, mean) observed for the full model and after removing the $\alpha$-scaling mechanism across the 102 UCR datasets. Disabling the $\alpha$-scaling mechanism leads to a notable drop in performance, from 2.1% to 0.09% with ASCENSION$_{\text{ResNet-Emb}}$, and from 1.2% to 0.4% with ASCENSION$_{\text{FCN-Emb}}$.

**Clustering loss ablation** Table 4 shows accuracy gains (Q1, median, Q3, mean) for the full model and after removing the clustering loss on 102 UCR datasets. Without this loss, ASCENSION$_{\text{ResNet-Emb}}$ drops from 2.1% to $-0.05\%$, and ASCENSION$_{\text{FCN-Emb}}$ from 1.2% to $-0.05\%$. Overlapping class distributions hinder discrimination, a known effect in clustering-based methods like VaDE. As seen in Figure 4(a), performance also declines progressively with each extra augmentation step.

Table 4: Average accuracy gains for the full model (**Full**, no component removed) and after individually removing $\alpha$-scaling or the clustering loss. Removing $\alpha$-scaling yields at most a marginal gain ($\leq 0.4\%$). Removing the clustering loss yields a $-0.05\%$ drop, confirming both components are essential for ASCENSION's improvements.

| Component removed | Classifier | ↑Q1 | ↑Median | ↑Q3 | ↑Mean |
|---|---|---|---|---|---|
| $\alpha$-scaling mechanism | ASCENSION$_{\text{ResNet-Emb}}$ | $-1.0\%$ | 0.2% | 1.5% | 0.09% |
| | ASCENSION$_{\text{FCN-Emb}}$ | $-0.4\%$ | 0.5% | 2% | 0.4% |
| Clustering loss | ASCENSION$_{\text{ResNet-Emb}}$ | $-1.5\%$ | $-0.8\%$ | 0.05% | $-0.05\%$ |
| | ASCENSION$_{\text{FCN-Emb}}$ | $-1.6\%$ | $-0.04\%$ | 0.05% | $-0.05\%$ |
| Full (no removal) | ASCENSION$_{\text{ResNet-Emb}}$ | 0.03% | 1.6% | 3.6% | **2.1%** |
| | ASCENSION$_{\text{FCN-Emb}}$ | $-0.02\%$ | 0.05% | 2% | **1.2%** |

**Loss weight hyperparameter tuning** Figure 4(b) reports the average accuracy variance over all loss-weight combinations examined for each dataset. Across most datasets, accuracy remains stable regardless

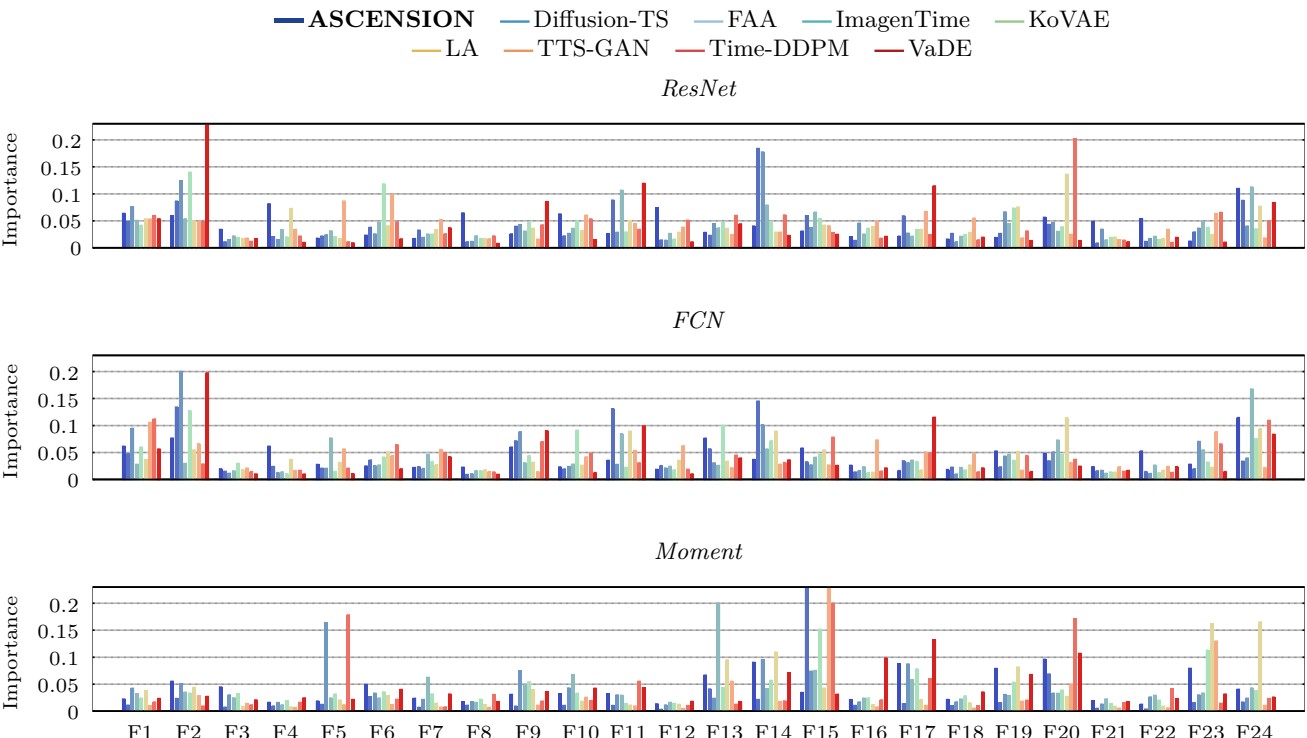

Figure 5: Feature importance over 24 features (F1–F24, see Appendix L). Key ones include F2: top $z$-score range, F14: mean time between extreme events, F20: slow fluctuations, F23: train/test size ratio, F24: train–test distribution discrepancy (Appendix M).

of the specific choice of $(\gamma_1, \gamma_2, \gamma_3, \gamma_4)$. Consequently, no parameter tuning seems necessary and setting all weights uniformly $(\gamma_1 = \gamma_2 = \gamma_3 = \gamma_4 = 1)$ is a reasonable choice.

### 4.3 Ablation 2: Feature-based gain analysis

As shown in Section 4.1.2, ASCENSION outperforms traditional and generative DA methods on most datasets, but 30–50% show no gain or even degradation (Unchanged/Worsened in Table 1; full results in Appendix J). To pinpoint which datasets benefit from DA, we perform a feature-based analysis using ASCENSION with ResNet, FCN, and Moment as external classifiers for fair comparison.

**Feature extraction**  We use the 22 **CA**nonical **T**ime-series **CH**aracteristics (`catch22`, Lubba et al., 2019) to featurise TS, and add two additional features: (F23) the train/test split ratio, and (F24) the distribution discrepancy ratio between train and test sets (see metric definitions in Appendix L). A full description of the 24 features (F1–F24) is provided in Appendix J. F23 is computable from dataset sizes and F24 requires the test set and is used here as a post-hoc indicator.

**Analysis methodology**  We average TS features per dataset to identify those most and least responsive to augmentation. We then assess how DA affects classification to pinpoint key influencing features. A shallow random forest with a high number of estimators is used to predict augmentation outcomes from the averaged F1–F24 features across DA methods. We repeat the protocol for ResNet, FCN, and Moment.

**Results**  Figure 5 shows that each DA method aligns with specific dataset features depending on the classifier. For ResNet/FCN, VaDE links to F2 (high $z$-score frequency), F11 (2D distance change), and F17 (longest decreasing trend). Diffusion-TS to F14 (rapid fluctuations); Time-DDPM (ResNet) to F20 (slow fluctuations); and ASCENSION to F24 (train–test distribution gap). With Moment, feature rankings

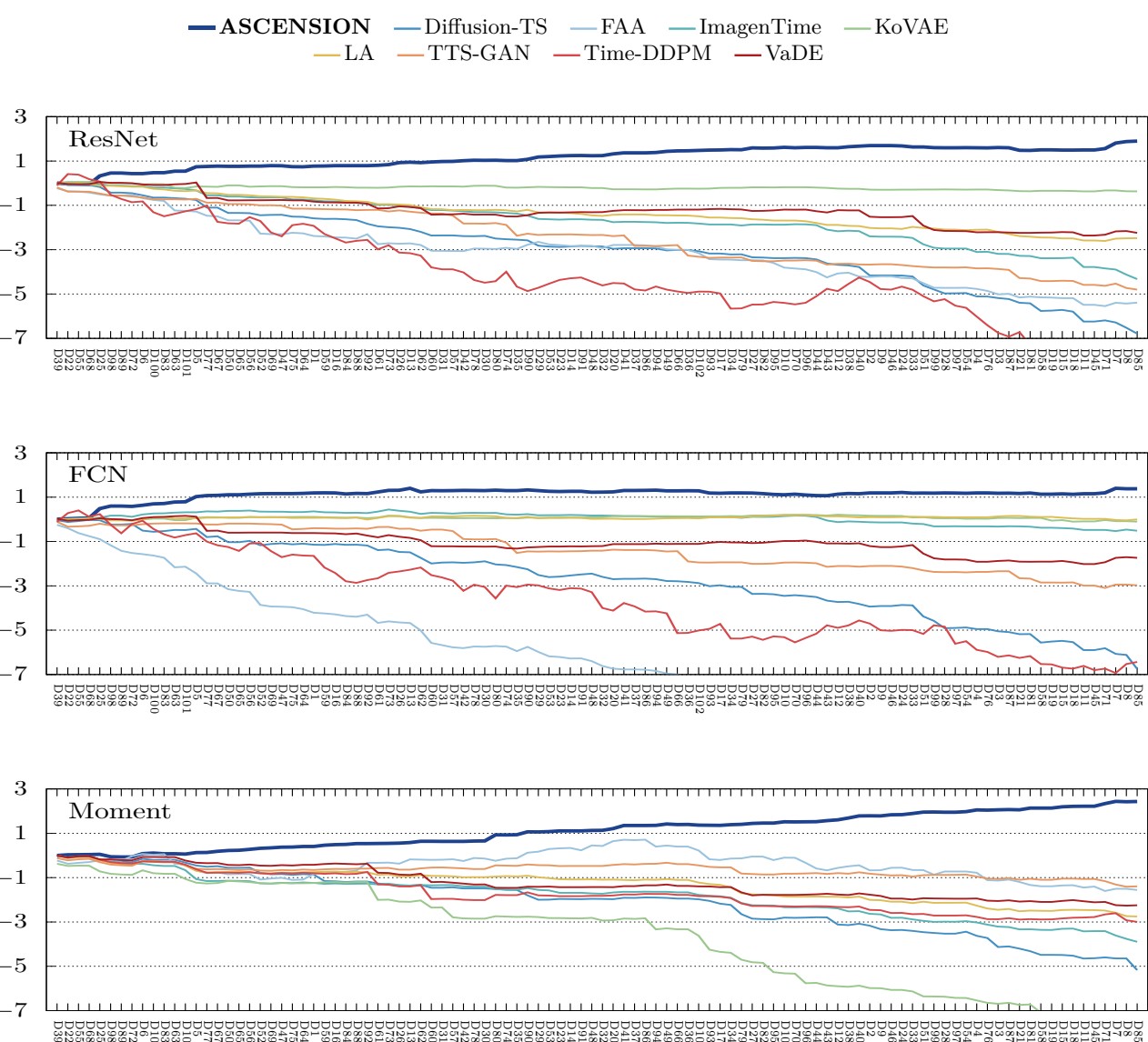

Figure 6: Cumulative accuracy improvement (%) vs. F24 (train–test distribution discrepancy; see Table A). Datasets are sorted by increasing discrepancy.

shift: F15 (rapid fluctuations) peaks sharply (0.64 for Diffusion-TS, 0.54 for TTS-GAN), while ASCENSION depends less on F24, and VaDE more, showing that pretraining reshapes which dataset traits drive augmentation gains. To investigate the discrepancy axis directly, we sort the 102 UCR datasets by ascending discrepancy (F24, Appendix M) and report the cumulative mean accuracy gain as datasets are added in this order (Figure 6).

ASCENSION is the only DA method whose cumulative gain stays positive at every cutoff (+2.78%, +1.88%, +1.87%, +1.67% at the 25%, 50%, 75%, 100% cumulative cutoffs on ResNet, with similar shapes on FCN and Moment), while every generative baseline stays below zero throughout. Crucially, the gap between ASCENSION and the strongest generative baseline (LA) does not close on the high-discrepancy datasets, *i.e.*, where in-distribution generative augmentation cannot help. This robustness across the full discrepancy range, not concentration in any one quartile, validates our central hypothesis: progressive expansion into

under-represented latent regions yields beneficial augmentations *regardless* of whether the test distribution differs from training.

## 5    Conclusion

This paper presents ASCENSION, a novel DA method built on the intuition that gradually exploring under-represented latent regions and expanding the initial data distribution can enhance DA effectiveness. ASCENSION leverages a VAE with a contrastive clustering loss to promote intra-class compactness and inter-class separability, and introduces an $\alpha$-scaling mechanism that progressively expands class densities to generate diverse, class-consistent samples. Across 102 UCR benchmarks and three classifiers (ResNet, FCN, Moment), ASCENSION delivers non-negative gains on 73.5% of dataset–classifier pairs, compared to 50.0% for the next-best baseline (VaDE), and a mean accuracy gain of roughly 2%, whereas every generative baseline we benchmark loses accuracy on average. An ablation study confirms that the $\alpha$-scaling mechanism is the single most impactful component of the method. These results position ASCENSION as the only DA method in our benchmark to offer consistent gains across the heterogeneous domains of the UCR archive, without requiring prior knowledge of method suitability.

## 6    Limitations and Future Work

Despite its contributions, this study has two limitations. First, our evaluation is restricted to *univariate* TS; extending the contrastive clustering loss to a joint-channel latent space, where inter-channel dependencies are modelled directly, is the natural next step toward multivariate sensor data. Second, the expansion uses a single scalar $\alpha$ applied uniformly across all classes. Alternative expansion mechanisms that adapt to class-specific covariance structure would relax this restriction and may further improve gains on heterogeneous dataset families.

## Broader Impact Statement

We evaluated ASCENSION on the publicly available UCR Time Series Classification archive (102 datasets spanning sensor, motion, image-derived, biomedical, and spectral domains) and identified no significant ethical concerns in its use. By delivering consistent classification gains across these heterogeneous domains without requiring per-dataset method selection, ASCENSION lowers the cost of deploying TS classifiers when labelled data is scarce. The generated samples remain class-consistent by construction (via the rejection step), which is especially valuable in scientific and operational settings where labels are sensitive or costly to obtain.

## Reproducibility Statement

We detailed exact implementation, hyperparameters, and provide code to produce our results at `https://anonymous.4open.science/r/ASCENSION_TMLR`. All experiments were run on a system equipped with an NVIDIA RTX 4080 Super GPU and an AMD Ryzen 7 7800X3D CPU at 4.2 GHz.

The 102 UCR datasets used in the experiments are publicly available in the UCR Time Series Classification Archive (Dau et al., 2019) and are listed individually in Appendix A. Training protocols, hyperparameter grids, and per-dataset results are reported in Appendices J and K.

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

# A  Dataset list

Table 5 lists the 102 UCR datasets used in our experiments and their D-id labels.

Table 5: Mapping from D-id to UCR dataset name and raw UCR domain. Datasets are sorted alphabetically; the same numbering is used throughout the paper (Tables 20–22 and 23). The macro-domain grouping (Industrial, Image, Motion, Spectral, Biomedical) is defined in Appendix B.

| # | Dataset | Domain | # | Dataset | Domain |
|---|---|---|---|---|---|
| D1 | ACSF1 | Sensor | D52 | Meat | Spectro |
| D2 | Adiac | Image | D53 | MedicalImages | Image |
| D3 | ArrowHead | Image | D54 | MiddlePhalanxOutlineAgeGroup | Image |
| D4 | BME | Simulated | D55 | MiddlePhalanxOutlineCorrect | Image |
| D5 | Beef | Spectro | D56 | MiddlePhalanxTW | Image |
| D6 | BeetleFly | Image | D57 | MixedShapesRegularTrain | Image |
| D7 | BirdChicken | Image | D58 | MixedShapesSmallTrain | Image |
| D8 | Car | Sensor | D59 | MoteStrain | Sensor |
| D9 | CBF | Simulated | D60 | NonInvasiveFetalECGThorax1 | ECG |
| D10 | ChlorineConcentration | Sensor | D61 | NonInvasiveFetalECGThorax2 | ECG |
| D11 | CinCECGTorso | ECG | D62 | OSULeaf | Image |
| D12 | Coffee | Spectro | D63 | OliveOil | Spectro |
| D13 | Computers | Device | D64 | PhalangesOutlinesCorrect | Image |
| D14 | Crop | Image | D65 | Plane | Sensor |
| D15 | DistalPhalanxOutlineAgeGroup | Image | D66 | PowerCons | Power |
| D16 | DistalPhalanxOutlineCorrect | Image | D67 | ProximalPhalanxOutlineAgeGroup | Image |
| D17 | DistalPhalanxTW | Image | D68 | ProximalPhalanxOutlineCorrect | Image |
| D18 | ECG200 | ECG | D69 | ProximalPhalanxTW | Image |
| D19 | ECG5000 | ECG | D70 | RefrigerationDevices | Device |
| D20 | ECGFiveDays | ECG | D71 | Rock | Spectrum |
| D21 | EOGHorizontalSignal | EOG | D72 | ScreenType | Device |
| D22 | EOGVerticalSignal | EOG | D73 | SemgHandGenderCh2 | Spectrum |
| D23 | Earthquakes | Sensor | D74 | SemgHandMovementCh2 | Spectrum |
| D24 | ElectricDevices | Device | D75 | SemgHandSubjectCh2 | Spectrum |
| D25 | EthanolLevel | Spectro | D76 | ShapeletSim | Simulated |
| D26 | FaceAll | Image | D77 | ShapesAll | Image |
| D27 | FaceFour | Image | D78 | SmallKitchenAppliances | Device |
| D28 | FacesUCR | Image | D79 | SmoothSubspace | Simulated |
| D29 | Fish | Image | D80 | SonyAIBORobotSurface1 | Sensor |
| D30 | FordA | Sensor | D81 | SonyAIBORobotSurface2 | Sensor |
| D31 | FordB | Sensor | D82 | StarLightCurves | Sensor |
| D32 | FreezerRegularTrain | Device | D83 | Strawberry | Spectro |
| D33 | FreezerSmallTrain | Device | D84 | SwedishLeaf | Image |
| D34 | GunPoint | Motion | D85 | Symbols | Image |
| D35 | GunPointAgeSpan | Motion | D86 | SyntheticControl | Simulated |
| D36 | GunPointMaleVersusFemale | Motion | D87 | ToeSegmentation1 | Motion |
| D37 | GunPointOldVersusYoung | Motion | D88 | ToeSegmentation2 | Motion |
| D38 | Ham | Spectro | D89 | Trace | Sensor |
| D39 | HandOutlines | Image | D90 | TwoLeadECG | Sensor |
| D40 | Haptics | Motion | D91 | TwoPatterns | Simulated |
| D41 | Herring | Image | D92 | UMD | Simulated |
| D42 | HouseTwenty | Sensor | D93 | UWaveGestureLibraryAll | Motion |
| D43 | InlineSkate | Motion | D94 | UWaveGestureLibraryX | Motion |
| D44 | InsectWingbeatSound | Sensor | D95 | UWaveGestureLibraryY | Motion |
| D45 | InsectEPGRegularTrain | EPG | D96 | UWaveGestureLibraryZ | Motion |
| D46 | InsectEPGSmallTrain | EPG | D97 | Wafer | Sensor |
| D47 | ItalyPowerDemand | Sensor | D98 | Wine | Spectro |
| D48 | LargeKitchenAppliances | Device | D99 | WordSynonyms | Image |
| D49 | Lightning2 | Sensor | D100 | Worms | Motion |
| D50 | Lightning7 | Sensor | D101 | WormsTwoClass | Motion |
| D51 | Mallat | Simulated | D102 | Yoga | Image |

## B Domain coverage of the UCR datasets

The 102 datasets span 11 UCR domain categories, visualised in Figure 7. The raw counts are Image (30), Sensor (19), Motion (14), Spectro (8), Simulated (8), Device (8), ECG (6), Spectrum (4), EPG (2), EOG (2), Power (1). For the macro-domain breakdown in Tables 16–18 we additionally group these 11 categories into five macro-domains: Industrial (35.3%), Image (29.4%), Motion (13.7%), Spectral (11.8%) and Biomedical (9.8%). All tasks are closed-set, supervised, fixed-label TSC (not anomaly detection, forecasting, or risk assessment).

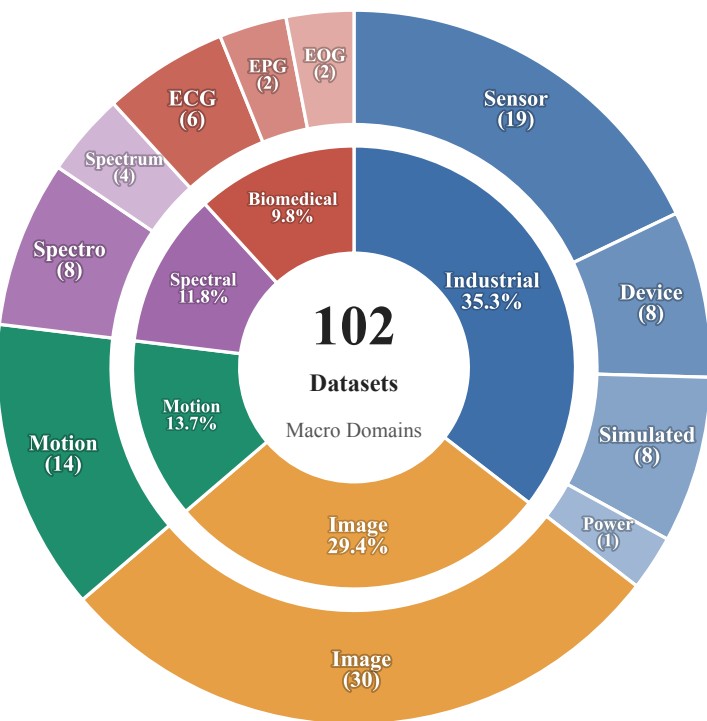

Figure 7: Domain coverage of the 102 UCR datasets used in our experiments. Inner ring: the five macro-domains (true percentages shown). Outer ring: the 11 raw UCR categories with absolute counts.

## C  Proof of Proposition 1

Let $\mu_i = \mu_\phi(x_i)$ and $\Sigma_i = \Sigma_\phi(x_i)$ and $\bar{\mu}_y = \frac{1}{|\mathcal{I}_y|} \sum_{i \in \mathcal{I}_y} \mu_i$ for compactness, we have:

$$\mathrm{Cov}_{Z \sim d_y^\alpha}[Z] = \mathbb{E}[Z\,Z^\top] - \mathbb{E}[Z]\,\mathbb{E}[Z]^\top$$

$$= \int z\,z^\top\,d_y^\alpha(z)\,\mathrm{d}z \;-\; \left( \int z\,d_y^\alpha(z)\,\mathrm{d}z \right)\left( \int z\,d_y^\alpha(z)\,\mathrm{d}z \right)^\top$$

$$= \int z\,z^\top\,\frac{1}{|\mathcal{I}_y|}\sum_{i \in \mathcal{I}_y} \mathcal{N}(z; \mu_i, \alpha\Sigma_i)\,\mathrm{d}z \;-\; \left( \int z\,\frac{1}{|\mathcal{I}_y|}\sum_{i \in \mathcal{I}_y} \mathcal{N}(z; \mu_i, \alpha\Sigma_i)\,\mathrm{d}z \right)\left( \int z\,\frac{1}{|\mathcal{I}_y|}\sum_{i \in \mathcal{I}_y} \mathcal{N}(z; \mu_i, \alpha\Sigma_i)\,\mathrm{d}z \right)^\top$$

$$= \frac{1}{|\mathcal{I}_y|}\sum_{i \in \mathcal{I}_y} \int z\,z^\top\,\mathcal{N}(z; \mu_i, \alpha\Sigma_i)\,\mathrm{d}z \;-\; \left( \frac{1}{|\mathcal{I}_y|}\sum_{i \in \mathcal{I}_y} \int z\,\mathcal{N}(z; \mu_i, \alpha\Sigma_i)\,\mathrm{d}z \right)\left( \frac{1}{|\mathcal{I}_y|}\sum_{i \in \mathcal{I}_y} \int z\,\mathcal{N}(z; \mu_i, \alpha\Sigma_i)\,\mathrm{d}z \right)^\top$$

$$= \frac{1}{|\mathcal{I}_y|}\sum_{i \in \mathcal{I}_y} \left[ \alpha\Sigma_i + \mu_i\mu_i^\top \right] \;-\; \left( \frac{1}{|\mathcal{I}_y|}\sum_{i \in \mathcal{I}_y} \mu_i \right)\left( \frac{1}{|\mathcal{I}_y|}\sum_{i \in \mathcal{I}_y} \mu_i \right)^\top$$

$$= \frac{1}{|\mathcal{I}_y|}\sum_{i \in \mathcal{I}_y} \left[ \alpha\Sigma_i + \mu_i\mu_i^\top \right] \;-\; \bar{\mu}_y\,\bar{\mu}_y^\top$$

$$= \alpha\,\frac{1}{|\mathcal{I}_y|}\sum_{i \in \mathcal{I}_y} \Sigma_i \;+\; \frac{1}{|\mathcal{I}_y|}\sum_{i \in \mathcal{I}_y} \mu_i\,\mu_i^\top \;-\; \bar{\mu}_y\,\bar{\mu}_y^\top$$

$$= \alpha\,\frac{1}{|\mathcal{I}_y|}\sum_{i \in \mathcal{I}_y} \Sigma_i \;+\; \frac{1}{|\mathcal{I}_y|}\sum_{i \in \mathcal{I}_y} \left( \mu_i - \bar{\mu}_y \right)\left( \mu_i - \bar{\mu}_y \right)^\top$$

$$= \alpha\,\bar{\Sigma}_w(y) + \Sigma_b(y).$$

$\square$

## D  Comparison with class-conditional VAE baselines

A natural alternative to ASCENSION is a class-conditional VAE (cVAE) that concatenates the class label to both encoder and decoder inputs. To isolate the effect of ASCENSION's contrastive clustering loss from the effect of class conditioning, we evaluate three cVAE variants with ResNet and FCN on all 102 UCR datasets: (i) the standard cVAE formulation with a fixed prior $z \sim \mathcal{N}(0, I)$; (ii) a prior-scaled variant $z \sim \mathcal{N}(0, \alpha I)$ with $\alpha > 1$ ($\alpha = 2.6$); and (iii) cVAE generations refined with ASCENSION's posterior-mixture sampling and filtering. The single $\alpha = 2.6$ matches the value used for wall-clock comparisons in Appendix I, the mean of the per-dataset best $\alpha$ values across the 102 UCR datasets (Tables 20-22). Variant (i) provides the no-scaling endpoint ($\alpha = 1$); variant (iii) uses ASCENSION's per-dataset $\alpha$ values and therefore covers the same $\alpha$ range as the headline comparison (Table 1).

Table 6: ASCENSION vs. three cVAE variants (no contrastive loss) on 102 UCR datasets.

| | DA method | Improved | | Unchanged | | Worsened | | Total | |
|---|---|---|---|---|---|---|---|---|---|
| | | $\uparrow\text{Nb}_{\text{data}}$ | $\uparrow\overline{\Delta\text{Acc}}$ | $\text{Nb}_{\text{data}}$ | $\overline{\Delta\text{Acc}}$ | $\downarrow\text{Nb}_{\text{data}}$ | $\downarrow\overline{\Delta\text{Acc}}$ | $\text{Nb}_{\text{data}}$ | $\overline{\Delta\text{Acc}}$ |
| ResNet | cVAE prior, $\mathcal{N}(0, I)$ | 32 | 3.9% | 17 | 0% | 53 | -2.3% | 102 | +0.05% |
| | cVAE prior, $\mathcal{N}(0, 2.6\,I)$ | 36 | 1.8% | 12 | 0% | 54 | -2.7% | 102 | $-0.76\%$ |
| | cVAE + posterior mixture | 42 | 2.0% | 15 | 0% | 45 | -1.7% | 102 | +0.09% |
| | **ASCENSION** | **59** | **4.1%** | 16 | 0% | **27** | **-1.9%** | 102 | **+1.9%** |
| FCN | cVAE prior, $\mathcal{N}(0, I)$ | 32 | 2.5% | 15 | 0% | 55 | -2.7% | 102 | $-0.68\%$ |
| | cVAE prior, $\mathcal{N}(0, 2.6\,I)$ | 25 | 2.5% | 13 | 0% | 64 | -3.2% | 102 | $-1.38\%$ |
| | cVAE + posterior mixture | 42 | 1.9% | 14 | 0% | 46 | -2.5% | 102 | $-0.32\%$ |
| | **ASCENSION** | **56** | **4.0%** | 12 | 0% | **34** | **-2.5%** | 102 | **+1.4%** |

**Why label conditioning is not enough?** The cVAE benefits from label conditioning, but frequently produces overlapping latent regions for classes with similar temporal patterns, which leads to degraded performance on roughly half of the datasets across all three variants (53/54/45 with ResNet and 55/64/46 with FCN, for the standard prior, prior-scaled, and posterior-mixture variants respectively). ASCENSION's margin-based contrastive loss actively pushes classes apart, enlarging the Bayes regions $\mathcal{R}_y^\alpha$ (Definition 2) that the $\alpha$-scaling mechanism populates. The contrastive loss is therefore not merely an alternative to label conditioning but an essential component of the method (cf Proposition 1 and Section 4.2, Table 4, where removing $\mathcal{L}_{\text{cluster}}$ collapses the mean augmentation gain from +2.1% to $-0.05\%$).

## E  Single-configuration reference results

For completeness, Table 7 reports the same comparison as Table 1 but with each method run under *a single fixed hyperparameter configuration* ($\alpha = 1.5$, $\gamma_1 = \gamma_2 = \gamma_3 = \gamma_4 = 1$) applied uniformly across all 102 datasets and all three classifiers.

Table 7: Reference Table 1 with a single fixed configuration per method, applied uniformly across all 102 UCR datasets and all three classifiers (no per-dataset hyperparameter selection).

| | DA method | (Venue, Year) | Improved | | Unchanged | | Worsened | | Total | |
|---|---|---|---|---|---|---|---|---|---|---|
| | | | $\uparrow$Nb$_{data}$ | $\uparrow$ $\overline{\Delta Acc}$ | Nb$_{data}$ | $\overline{\Delta Acc}$ | $\downarrow$Nb$_{data}$ | $\downarrow$ $\overline{\Delta Acc}$ | Nb$_{data}$ | $\overline{\Delta Acc}$ |
| ResNet | VaDE | (IJCAI, 2016) | 44 | 2.8% | 9 | 0% | 49 | -7.1% | 102 | $-2.2\%$ |
| | FAA | (NeurIPS, 2019) | 19 | 7.8% | 8 | 0% | 75 | -9.2% | 102 | $-5.3\%$ |
| | TTS-GAN | (AIME, 2022) | 26 | 2.2% | 11 | 0% | 65 | -8.3% | 102 | $-4.7\%$ |
| | KoVAE | (ICLR, 2024) | 38 | 1.5% | 19 | 0% | 45 | -2.1% | 102 | $-0.4\%$ |
| | Time-DDPM | (SENS. J., 2023) | 38 | **16.3%** | 0 | 0% | 64 | -22.3% | 102 | $-7.9\%$ |
| | LA | (ArXiv, 2023) | 14 | 3.3% | 13 | 0% | 75 | -3.9% | 102 | $-2.4\%$ |
| | ImagenTime | (NeurIPS, 2024) | 15 | 1.3% | 13 | 0% | 74 | -6.1% | 102 | $-4.2\%$ |
| | Diffusion-TS | (ICLR, 2024) | 21 | 1.3% | 5 | 0% | 76 | -9.3% | 102 | $-6.7\%$ |
| | **ASCENSION** | | **56** | 4.0% | 16 | 0% | **30** | **-1.7%** | 102 | **+1.7%** |
| FCN | VaDE | (IJCAI, 2016) | 42 | 3.4% | 15 | 0% | 45 | -7.0% | 102 | $-1.7\%$ |
| | FAA | (NeurIPS, 2019) | 14 | 7.1% | 3 | 0% | 85 | -15.0% | 102 | $-11.5\%$ |
| | TTS-GAN | (AIME, 2022) | 36 | 2.9% | 17 | 0% | 49 | -8.2% | 102 | $-2.9\%$ |
| | KoVAE | (ICLR, 2024) | 41 | 2.1% | 15 | 0% | 46 | **-2.1%** | 102 | $-0.1\%$ |
| | Time-DDPM | (SENS. J., 2023) | 42 | **16.9%** | 1 | 0% | 59 | -23.0% | 102 | $-6.3\%$ |
| | LA | (ArXiv, 2023) | 44 | 2.3% | 19 | 0% | 39 | -2.6% | 102 | $0.0\%$ |
| | ImagenTime | (NeurIPS, 2024) | 38 | 2.7% | 11 | 0% | 53 | -2.9% | 102 | $-0.5\%$ |
| | Diffusion-TS | (ICLR, 2024) | 36 | 3.3% | 7 | 0% | 59 | -13.5% | 102 | $-6.6\%$ |
| | **ASCENSION** | | **50** | 3.0% | 13 | 0% | **39** | **-1.4%** | 102 | **+1.0%** |
| Moment | VaDE | (IJCAI, 2016) | 35 | 2.9% | 8 | 0% | 59 | -5.6% | 102 | $-2.2\%$ |
| | FAA | (NeurIPS, 2019) | 50 | **7.8%** | 6 | 0% | 46 | -11.8% | 102 | $-1.5\%$ |
| | TTS-GAN | (AIME, 2022) | 41 | 3.4% | 7 | 0% | 54 | -5.2% | 102 | $-1.4\%$ |
| | KoVAE | (ICLR, 2024) | 27 | 3.7% | 5 | 0% | 70 | -12.2% | 102 | $-7.4\%$ |
| | Time-DDPM | (SENS. J., 2023) | 35 | 3.8% | 5 | 0% | 62 | -7.0% | 102 | $-2.9\%$ |
| | LA | (ArXiv, 2023) | 31 | 2.6% | 6 | 0% | 65 | -5.5% | 102 | $-2.7\%$ |
| | ImagenTime | (NeurIPS, 2024) | 25 | 2.1% | 7 | 0% | 70 | -6.3% | 102 | $-3.8\%$ |
| | Diffusion-TS | (ICLR, 2024) | 28 | 3.2% | 5 | 0% | 69 | -8.8% | 102 | $-5.1\%$ |
| | **ASCENSION** | | **48** | 3.4% | 9 | 0% | 45 | **-2.2%** | 102 | **+0.6%** |

Under the single-configuration setting, ASCENSION still posts the only positive total on every classifier ($+1.7\%, +1.0\%, +0.6\%$), the highest improved-dataset count, and the lowest degradation count on ResNet/FCN. Per-dataset hyperparameter selection (Table 1) widens these margins on ResNet and Moment ($+1.9\%, +2.4\%$) and improves several competitor methods correspondingly.

# F   Statistical significance of headline accuracy results

To complement Table 1, we report Friedman tests and critical-difference (CD) diagrams (Nemenyi post-hoc at the 5% significance level) following (Demšar, 2006), computed over per-dataset accuracies for each of the three classifiers. The Friedman test rejects the null hypothesis that all DA methods perform equally with $p < 10^{-18}$ on every classifier; the critical-distance threshold for $k = 9$ methods and $N = 102$ datasets is CD $\approx 1.19$. Figures 8, 9, and 10 show the resulting CD diagrams for ResNet, FCN, and Moment respectively.

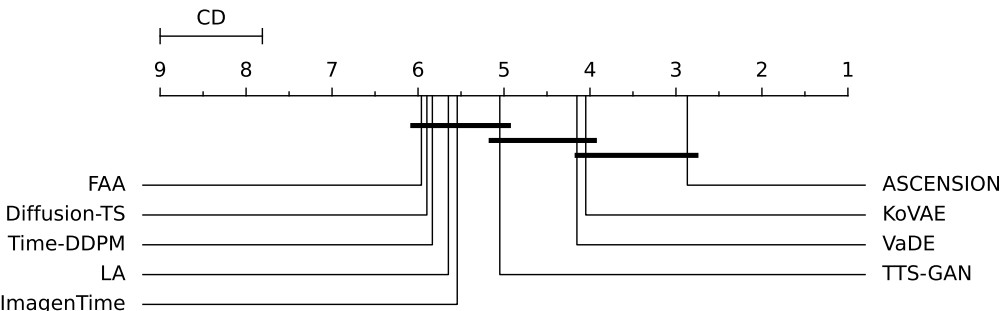

Figure 8: Critical-difference diagram, ResNet classifier (Friedman + Nemenyi at the 5% significance level).

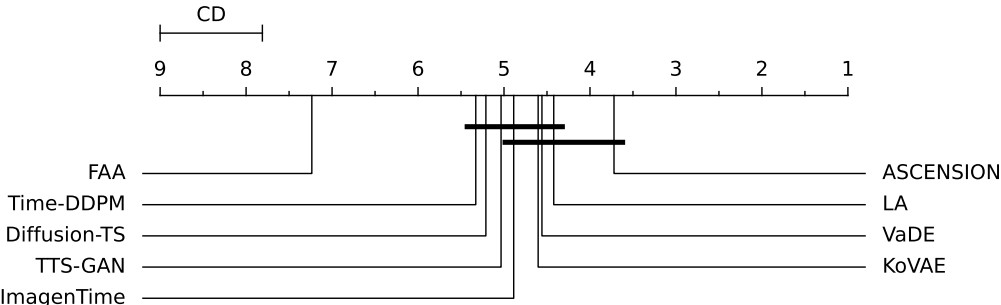

Figure 9: Critical-difference diagram, FCN classifier (Friedman + Nemenyi at the 5% significance level).

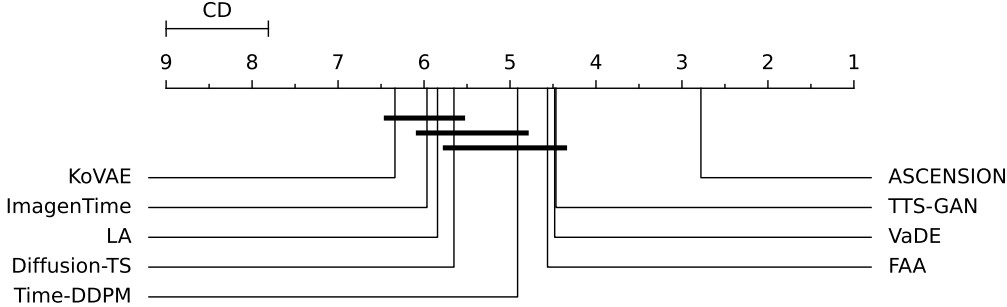

Figure 10: Critical-difference diagram, Moment classifier (Friedman + Nemenyi at the 5% significance level).

ASCENSION achieves the lowest mean rank on every classifier (2.87, 3.72, 2.79 for ResNet, FCN, Moment). On the Moment classifier the Nemenyi post-hoc places ASCENSION significantly ahead of every other DA method. On ResNet, ASCENSION is significantly better than all baselines except KoVAE, whose rank gap to ASCENSION (1.18) is just inside the critical distance. On FCN, ASCENSION is significantly better than FAA, Time-DDPM, Diffusion-TS, and TTS-GAN; the remaining baselines (KoVAE, LA, VaDE, ImagenTime) occupy a shared rank band with ASCENSION at its lower edge.

## G Empirical acceptance rate of the rejection step

As a robustness check we measure, for each augmentation iteration $k$, the empirical acceptance rate

$$\mathcal{C}_y^k = \mathbb{P}_{Z \sim d_y^{\alpha^k}} \left[ \arg\max_{y'} d_{y'}^{\alpha^k}(Z) = y \right], \tag{8}$$

i.e. the fraction of $\alpha^k$-scaled samples that the likelihood-dominance rule classifies into the originating class $y$. This is the empirical counterpart of the acceptance rate plotted in Figure 3: same quantity, real encoded UCR data instead of an analytic Gaussian model. We estimate $\mathcal{C}_y^k$ by sampling $n = 1000$ latents per class.

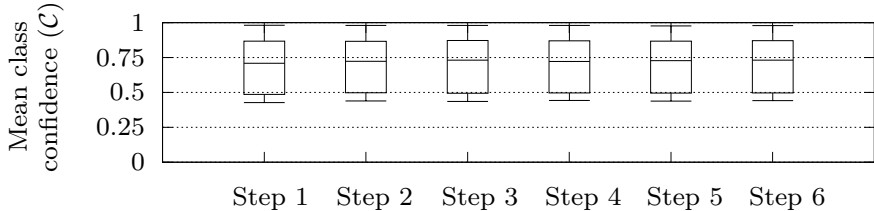

Figure 11: Empirical acceptance rate $\mathcal{C}$ across six expansion iterations on the 102 UCR datasets. Median $\mathcal{C}$ stays in the 0.65-0.70 band with $Q_1 \approx 0.5$ and $Q_3 \approx 0.8$ across iterations, and shows no systematic decline as $k$ grows: the rejection step retains the majority of candidates at every step.

The stability of the median across iterations is consistent with Proposition 1: the contrastive margin enforced by $\mathcal{L}_{\text{cluster}}$ keeps $m/\sigma_w$ above $2\sqrt{\alpha}$ across iterations, so the operating point stays in the class-preserving regime of Figure 3. At $\alpha \approx 2.6$, this places the operating point slightly above the bold curve $m/\sigma_w = 2\sqrt{\alpha} \approx 3.2$, exactly in the region where the theoretical heatmap predicts acceptance in the 0.65-0.7 range, matching the empirical distribution observed here.

## H Complexity

Let $N$ be the training-set size, $E$ the number of training epochs, $D$ the input dimension (TS length), $H$ the hidden width, $S$ the number of synthetic samples, and $T$ the number of outer iterations. Table 8 summarises the asymptotic training and generation costs by method family.

Table 8: Asymptotic training and generation complexity by method family. $T$ is the number of outer ASCENSION iterations or the number of denoising steps for diffusion methods (typically $T \approx 1000$).

| Method family | Training | Per-sample generation |
|---|---|---|
| FAA (transformation) | n/a | $\mathcal{O}(D)$ |
| VaDE / KoVAE | $\mathcal{O}(N \cdot E \cdot D \cdot H)$ | $\mathcal{O}(D \cdot H)$ |
| TS-GAN / LatentAugment | $\mathcal{O}(N \cdot E \cdot D \cdot H)$ | $\mathcal{O}(D \cdot H)$ |
| Diffusion-TS / ImagenTime | $\mathcal{O}(N \cdot E \cdot D \cdot H)$ | $\mathcal{O}(T \cdot D \cdot H)$ |
| **ASCENSION** | $\mathcal{O}(T_{\text{outer}} \cdot N \cdot E \cdot D \cdot H)$ | $\mathcal{O}(D \cdot H)$ |

At the canonical setting $T_{\text{outer}} = 1$, ASCENSION's training cost matches other VAE baselines. The crucial gap is at generation time: ASCENSION produces each synthetic instance with a single forward pass through the decoder, whereas diffusion baselines require $\sim 1000$ sequential denoising steps.

## I Wall-clock timing

Table 9 reports end-to-end wall-clock time (generative-model training plus sample generation; downstream classifier training is excluded as it is identical for every method) on three UCR datasets chosen to cover the

dataset-size range. ASCENSION is run with $\alpha = 2.6$, the mean of the per-dataset best $\alpha$ values across our sweep (Tables 20–22). All measurements on the same NVIDIA RTX 4080 Super used for the main-results experiments.

Table 9: Wall-clock time (seconds) for generative-model training plus generation, on three representative UCR datasets. Same hardware and software stack as the main experiments.

| Dataset | FAA | VaDE | Time-DDPM | TTS-GAN | **ASCENSION** | Diffusion-TS | ImagenTime |
|---|---|---|---|---|---|---|---|
| Coffee ($n\!=\!28$) | $<0.1$ | 0.8 | 0.6 | 4.2 | **1.0** | 13.2 | 20.3 |
| ECG200 ($n\!=\!100$) | $<0.1$ | 1.9 | 2.2 | 8.7 | **2.8** | 24.5 | 74.7 |
| FaceAll ($n\!=\!560$) | 0.01 | 5.8 | 5.7 | 25.8 | **11.0** | 89.1 | 188.9 |

ASCENSION is roughly an order of magnitude faster than Diffusion-TS ($\sim 9$–$13\times$) and $15$–$27\times$ faster than ImagenTime, both of which require hundreds of sequential denoising steps per sample. It is $2$–$4\times$ faster than TTS-GAN. It is in the same computational range as VaDE and Time-DDPM, though consistently a bit slower because it additionally computes the contrastive loss. FAA is near-instant as it applies fixed transformations with no training but does not offer downstream gain (Table 1).

## J    Detailed classification performance

This section analyses average classification accuracy changes across the 11 UCR domains (Table 19), with mean gains and drops reported in Tables 10–15 for ResNet, FCN, and Moment. Tables 16–18 aggregate these per-domain results into the five macro-domains defined in Appendix B.

Table 10: Mean improvement in classification accuracy across dataset domains (ResNet) best in bold.

| Type | ASCENSION | | Diffusion-TS | | FAA | | ImagenTime | | KoVAE | | LA | | Time-DDPM | | TTS-GAN | | VaDE | |
|---|---|---|---|---|---|---|---|---|---|---|---|---|---|---|---|---|---|---|
| | ↑Nb | ↑ $\overline{\Delta Acc}$ | ↑Nb | ↑ $\overline{\Delta Acc}$ | ↑Nb | ↑ $\overline{\Delta Acc}$ | ↑Nb | ↑ $\overline{\Delta Acc}$ | ↑Nb | ↑ $\overline{\Delta Acc}$ | ↑Nb | ↑ $\overline{\Delta Acc}$ | ↑Nb | ↑ $\overline{\Delta Acc}$ | ↑Nb | ↑ $\overline{\Delta Acc}$ | ↑Nb | ↑ $\overline{\Delta Acc}$ |
| Device | 3/8 | 1.4% | 1/8 | 1.3% | 4/8 | 4.4% | 1/8 | 1.9% | **5/8** | **1.0%** | 1/8 | 8.9% | 1/8 | 13.7% | 2/8 | 0.3% | 3/8 | 2.0% |
| Image | **22/30** | **3.7%** | 8/30 | 1.5% | 4/30 | 8.2% | 5/30 | 1.8% | 11/30 | 1.6% | 3/30 | 4.0% | 11/30 | 11.8% | 10/30 | 2.5% | 17/30 | 2.9% |
| Simulated | 1/8 | 0.6% | **4/8** | **0.8%** | 2/8 | 11.3% | 1/8 | 0.6% | 3/8 | 0.4% | 2/8 | 1.3% | 3/8 | 3.0% | **4/8** | 0.8% | 2/8 | 0.8% |
| Spectro | **5/8** | **16.4%** | 1/8 | 0.4% | 2/8 | 10.8% | 2/8 | 1.6% | 2/8 | 4.2% | 1/8 | 3.3% | 3/8 | 17.7% | 3/8 | 2.6% | 4/8 | 7.5% |
| Sensor | **12/19** | **2.8%** | 4/19 | 0.6% | 2/19 | 11.2% | 3/19 | 0.3% | 6/19 | 1.9% | 5/19 | 2.5% | 5/19 | 18.4% | 4/19 | 3.9% | 8/19 | 2.0% |
| ECG | **5/6** | **2.7%** | 1/6 | 1.2% | 1/6 | 13.8% | 1/6 | 2.0% | 2/6 | 0.7% | 1/6 | 5.5% | 2/6 | 7.7% | 1/6 | 1.0% | 2/6 | 2.9% |
| EOG | 0/2 | 0.0% | 0/2 | 0.0% | 0/2 | 0.0% | 0/2 | 0.0% | 1/2 | 4.4% | 1/2 | 1.1% | **2/2** | **35.7%** | 0/2 | 0.0% | 0/2 | 0.0% |
| Motion | **8/14** | **1.7%** | 0/14 | 0.0% | 1/14 | 2.2% | 0/14 | 0.0% | 5/14 | 0.5% | 0/14 | 0.0% | 7/14 | 17.0% | 1/14 | 1.1% | 5/14 | 1.4% |
| EPG | **0/2** | **0.0%** | **0/2** | **0.0%** | **0/2** | **0.0%** | **0/2** | **0.0%** | **0/2** | **0.0%** | **0/2** | **0.0%** | **0/2** | **0.0%** | **0/2** | **0.0%** | **0/2** | **0.0%** |
| Power | **1/1** | **2.2%** | 1/1 | 2.1% | 0/1 | 0.0% | **1/1** | 1.1% | **1/1** | 1.7% | 0/1 | 0.0% | 0/1 | 0.0% | **1/1** | **2.2%** | 1/1 | 1.7% |
| Spectrum | 2/4 | 4.4% | 1/4 | 4.9% | 3/4 | 5.1% | 1/4 | 0.7% | 2/4 | 2.6% | 0/4 | 0.0% | **4/4** | **29.1%** | 0/4 | 0.0% | 2/4 | 2.5% |

Table 11: Mean worsened accuracy change across dataset domains (ResNet). Smallest reductions in bold.

| Type | ASCENSION | | Diffusion-TS | | FAA | | ImagenTime | | KoVAE | | LA | | Time-DDPM | | TTS-GAN | | VaDE | |
|---|---|---|---|---|---|---|---|---|---|---|---|---|---|---|---|---|---|---|
| | ↓Nb | ↓ $\overline{\Delta Acc}$ | ↓Nb | ↓ $\overline{\Delta Acc}$ | ↓Nb | ↓ $\overline{\Delta Acc}$ | ↓Nb | ↓ $\overline{\Delta Acc}$ | ↓Nb | ↓ $\overline{\Delta Acc}$ | ↓Nb | ↓ $\overline{\Delta Acc}$ | ↓Nb | ↓ $\overline{\Delta Acc}$ | ↓Nb | ↓ $\overline{\Delta Acc}$ | ↓Nb | ↓ $\overline{\Delta Acc}$ |
| Device | 5/8 | -2.1% | 7/8 | -2.3% | 4/8 | -5.7% | 7/8 | -2.7% | **1/8** | **-3.2%** | 7/8 | -2.6% | 7/8 | -15.5% | 5/8 | -3.7% | 5/8 | -2.2% |
| Image | **7/30** | **-1.5%** | 22/30 | -11.9% | 24/30 | -6.7% | 21/30 | -7.3% | 15/30 | -1.7% | 24/30 | -4.4% | 19/30 | -21.8% | 18/30 | -6.1% | 12/30 | -12.6% |
| Simulated | 4/8 | -0.5% | 4/8 | -18.1% | 6/8 | -6.7% | 6/8 | -9.2% | **2/8** | **-0.4%** | 6/8 | -2.6% | 5/8 | -28.8% | 3/8 | -3.3% | 6/8 | -9.4% |
| Spectro | **0/8** | **0.0%** | 7/8 | -8.6% | 6/8 | -20.6% | 5/8 | -5.2% | 5/8 | -5.3% | 5/8 | -5.1% | 5/8 | -23.3% | 4/8 | -5.7% | 3/8 | -1.6% |
| Sensor | **5/19** | **-0.7%** | 13/19 | -7.3% | 15/19 | -6.0% | 13/19 | -4.6% | 9/19 | -1.0% | 11/19 | -2.8% | 14/19 | -18.0% | 12/19 | -6.1% | 8/19 | -3.5% |
| ECG | **0/6** | **0.0%** | 5/6 | -18.3% | 4/6 | -25.3% | 5/6 | -13.1% | 4/6 | -2.2% | 4/6 | -7.2% | 4/6 | -36.0% | 5/6 | -5.4% | 4/6 | -16.6% |
| EOG | 2/2 | -7.3% | 2/2 | -12.4% | 2/2 | -15.1% | 2/2 | -4.7% | 1/2 | -2.8% | 1/2 | -9.4% | **0/2** | **0.0%** | 2/2 | -27.8% | 2/2 | -3.0% |
| Motion | **2/14** | **-1.6%** | 13/14 | -5.3% | 12/14 | -9.1% | 12/14 | -3.9% | 6/14 | -1.2% | 13/14 | -4.2% | 7/14 | -33.7% | 12/14 | -15.6% | 7/14 | -2.0% |
| EPG | **0/2** | **0.0%** | **0/2** | **0.0%** | **0/2** | **0.0%** | **0/2** | **0.0%** | **0/2** | **0.0%** | **0/2** | **0.0%** | 2/2 | -2.8% | **0/2** | **0.0%** | **0/2** | **0.0%** |
| Power | **0/1** | **0.0%** | **0/1** | **0.0%** | 1/1 | -2.2% | **0/1** | **0.0%** | **0/1** | **0.0%** | 1/1 | -1.1% | 1/1 | -8.5% | **0/1** | **0.0%** | **0/1** | **0.0%** |
| Spectrum | 2/4 | -2.9% | 3/4 | -5.0% | 1/4 | -6.0% | 3/4 | -5.9% | 2/4 | -6.1% | 3/4 | -1.7% | **0/4** | **0.0%** | 4/4 | -8.3% | 2/4 | -4.7% |

Table 12: Mean improvement in classification accuracy across dataset domains (FCN) best in bold.

| Type | ASCENSION ↑Nb | ↑ $\overline{\Delta Acc}$ | Diffusion-TS ↑Nb | ↑ $\overline{\Delta Acc}$ | FAA ↑Nb | ↑ $\overline{\Delta Acc}$ | ImagenTime ↑Nb | ↑ $\overline{\Delta Acc}$ | KoVAE ↑Nb | ↑ $\overline{\Delta Acc}$ | LA ↑Nb | ↑ $\overline{\Delta Acc}$ | Time-DDPM ↑Nb | ↑ $\overline{\Delta Acc}$ | TTS-GAN ↑Nb | ↑ $\overline{\Delta Acc}$ | VaDE ↑Nb | ↑ $\overline{\Delta Acc}$ |
|---|---|---|---|---|---|---|---|---|---|---|---|---|---|---|---|---|---|---|
| Device | 4/8 | 3.5% | **5/8** | **3.2%** | 1/8 | 7.2% | 4/8 | 0.7% | 4/8 | 0.9% | 4/8 | 1.0% | 4/8 | 20.2% | 3/8 | 0.9% | 4/8 | 2.7% |
| Image | **17/30** | **3.1%** | 13/30 | 3.8% | 2/30 | 7.3% | 9/30 | 2.9% | 12/30 | 2.3% | 14/30 | 2.1% | 13/30 | 17.0% | 11/30 | 3.2% | 15/30 | 3.5% |
| Simulated | **5/8** | **0.4%** | 3/8 | 2.8% | 3/8 | 3.6% | 0/8 | 0.0% | 3/8 | 0.6% | 0/8 | 0.0% | 1/8 | 12.6% | 4/8 | 0.9% | 0/8 | 0.0% |
| Spectro | **8/8** | **12.1%** | 4/8 | 3.6% | 2/8 | 11.3% | 5/8 | 3.7% | 2/8 | 6.8% | 2/8 | 1.4% | 4/8 | 8.3% | 3/8 | 4.8% | 3/8 | 3.7% |
| Sensor | 7/19 | 1.8% | 7/19 | 2.4% | 3/19 | 10.1% | **9/19** | **2.4%** | 6/19 | 1.2% | 8/19 | 1.9% | 4/19 | 19.9% | 6/19 | 3.4% | 7/19 | 2.1% |
| ECG | **5/6** | **3.5%** | 1/6 | 4.7% | 0/6 | 0.0% | 2/6 | 5.5% | 4/6 | 2.5% | 4/6 | 3.7% | 2/6 | 10.4% | 4/6 | 4.8% | 1/6 | 9.2% |
| EOG | **1/2** | 1.9% | 0/2 | 0.0% | 0/2 | 0.0% | **1/2** | 1.1% | 0/2 | 0.0% | **1/2** | 1.1% | **1/2** | **39.3%** | 0/2 | 0.0% | 0/2 | 0.0% |
| Motion | 7/14 | 2.1% | 1/14 | 0.5% | 2/14 | 3.3% | 6/14 | 1.8% | 6/14 | 1.3% | **9/14** | 2.9% | **9/14** | **13.6%** | 3/14 | 1.7% | **9/14** | 2.8% |
| EPG | **0/2** | **0.0%** | **0/2** | **0.0%** | **0/2** | **0.0%** | **0/2** | **0.0%** | **0/2** | **0.0%** | **0/2** | **0.0%** | **0/2** | **0.0%** | **0/2** | **0.0%** | **0/2** | **0.0%** |
| Power | 0/1 | 0.0% | 0/1 | 0.0% | 0/1 | 0.0% | **1/1** | **1.7%** | 0/1 | 0.0% | **1/1** | 0.6% | 0/1 | 0.0% | **1/1** | **1.7%** | 1/1 | 0.6% |
| Spectrum | 2/4 | 5.7% | 2/4 | 5.5% | 1/4 | 6.8% | 1/4 | 9.3% | **4/4** | 4.0% | 1/4 | 7.5% | **4/4** | **25.2%** | 1/4 | 0.7% | 2/4 | 8.9% |

Table 13: Mean worsened accuracy change across dataset domains (FCN). Smallest reductions in bold.

| Type | ASCENSION ↓Nb | ↓ $\overline{\Delta Acc}$ | Diffusion-TS ↓Nb | ↓ $\overline{\Delta Acc}$ | FAA ↓Nb | ↓ $\overline{\Delta Acc}$ | ImagenTime ↓Nb | ↓ $\overline{\Delta Acc}$ | KoVAE ↓Nb | ↓ $\overline{\Delta Acc}$ | LA ↓Nb | ↓ $\overline{\Delta Acc}$ | Time-DDPM ↓Nb | ↓ $\overline{\Delta Acc}$ | TTS-GAN ↓Nb | ↓ $\overline{\Delta Acc}$ | VaDE ↓Nb | ↓ $\overline{\Delta Acc}$ |
|---|---|---|---|---|---|---|---|---|---|---|---|---|---|---|---|---|---|---|
| Device | 4/8 | -2.2% | **3/8** | -5.8% | 7/8 | -9.0% | 4/8 | -4.2% | 4/8 | -3.7% | 4/8 | -3.2% | 4/8 | -27.9% | **3/8** | -3.8% | **3/8** | **-3.0%** |
| Image | **12/30** | -3.1% | 17/30 | -22.6% | 28/30 | -15.9% | 20/30 | -3.0% | 16/30 | -1.7% | **12/30** | **-2.0%** | 17/30 | -20.0% | 17/30 | -5.1% | 13/30 | -10.2% |
| Simulated | **1/8** | **-1.4%** | 3/8 | -20.3% | 5/8 | -18.2% | 6/8 | -1.6% | 5/8 | -1.0% | 4/8 | -2.8% | 6/8 | -20.2% | 2/8 | -5.8% | 6/8 | -10.4% |
| Spectro | **0/8** | **0.0%** | 4/8 | -11.7% | 6/8 | -30.4% | 2/8 | -6.1% | 4/8 | -4.3% | 4/8 | -4.9% | 4/8 | -24.5% | 3/8 | -3.2% | 3/8 | -2.7% |
| Sensor | 9/19 | -1.6% | 10/19 | -8.4% | 16/19 | -10.5% | **7/19** | -2.2% | **7/19** | **-1.1%** | 8/19 | -1.7% | 15/19 | -25.0% | 9/19 | -5.4% | 9/19 | -2.3% |
| ECG | **1/6** | **-2.0%** | 5/6 | -19.7% | 6/6 | -22.0% | 4/6 | -2.3% | 2/6 | -9.5% | 2/6 | -2.6% | 4/6 | -13.1% | 2/6 | -7.3% | 4/6 | -12.3% |
| EOG | 1/2 | -1.1% | 2/2 | -8.6% | 2/2 | -16.0% | 1/2 | -0.6% | 2/2 | -1.0% | **0/2** | **0.0%** | 1/2 | -11.3% | 2/2 | -27.1% | 2/2 | -6.2% |
| Motion | 5/14 | -3.8% | 12/14 | -6.4% | 11/14 | -12.7% | 7/14 | -3.1% | 5/14 | -1.0% | **3/14** | **-1.8%** | 5/14 | -26.4% | 8/14 | -14.5% | **3/14** | -3.9% |
| EPG | **0/2** | **0.0%** | **0/2** | **0.0%** | **0/2** | **0.0%** | **0/2** | **0.0%** | **0/2** | **0.0%** | **0/2** | **0.0%** | 2/2 | -11.1% | **0/2** | **0.0%** | **0/2** | **0.0%** |
| Power | **0/1** | **0.0%** | 1/1 | -1.1% | 1/1 | -3.9% | **0/1** | **0.0%** | 1/1 | -0.6% | **0/1** | **0.0%** | 1/1 | -89.6% | **0/1** | **0.0%** | **0/1** | **0.0%** |
| Spectrum | 1/4 | -2.0% | 2/4 | -4.2% | 3/4 | -6.1% | 2/4 | -4.2% | **0/4** | **0.0%** | 2/4 | -5.0% | **0/4** | **0.0%** | 3/4 | -15.9% | 2/4 | -4.7% |

Table 14: Mean improvement in classification accuracy across dataset domains (Moment) best in bold.

| Type | ASCENSION | | Diffusion-TS | | FAA | | ImagenTime | | KoVAE | | LA | | Time-DDPM | | TTS-GAN | | VaDE | |
|---|---|---|---|---|---|---|---|---|---|---|---|---|---|---|---|---|---|---|
| | ↑Nb | ↑ $\overline{\Delta Acc}$ | ↑Nb | ↑ $\overline{\Delta Acc}$ | ↑Nb | ↑ $\overline{\Delta Acc}$ | ↑Nb | ↑ $\overline{\Delta Acc}$ | ↑Nb | ↑ $\overline{\Delta Acc}$ | ↑Nb | ↑ $\overline{\Delta Acc}$ | ↑Nb | ↑ $\overline{\Delta Acc}$ | ↑Nb | ↑ $\overline{\Delta Acc}$ | ↑Nb | ↑ $\overline{\Delta Acc}$ |
| Device | **6/8** | **2.1%** | 3/8 | 1.5% | 5/8 | 15.1% | 1/8 | 0.1% | 1/8 | 2.4% | 2/8 | 0.7% | 2/8 | 0.9% | 2/8 | 0.5% | 1/8 | 0.5% |
| Image | **24/30** | **3.5%** | 11/30 | 4.5% | 16/30 | 5.8% | 9/30 | 2.8% | 9/30 | 4.8% | 10/30 | 3.6% | 12/30 | 3.8% | 14/30 | 3.9% | 12/30 | 3.8% |
| Simulated | **4/8** | 4.0% | 1/8 | 2.3% | **4/8** | **9.8%** | 2/8 | 0.8% | 1/8 | 2.7% | 2/8 | 4.5% | 3/8 | 4.1% | 3/8 | 3.3% | **4/8** | 3.1% |
| Spectro | **5/8** | **5.0%** | 1/8 | 1.1% | 4/8 | 7.1% | 0/8 | 0.0% | 0/8 | 0.0% | 0/8 | 0.0% | 0/8 | 0.0% | 1/8 | 3.6% | 0/8 | 0.0% |
| Sensor | **13/19** | **6.1%** | 5/19 | 2.7% | 9/19 | 9.1% | 6/19 | 2.2% | 10/19 | 3.8% | 9/19 | 2.3% | 8/19 | 5.4% | 11/19 | 4.0% | 9/19 | 2.6% |
| ECG | **4/6** | 2.7% | 1/6 | 0.4% | 3/6 | 7.6% | 2/6 | 3.3% | 2/6 | 0.9% | 2/6 | 0.7% | **4/6** | 1.4% | **4/6** | **2.9%** | 2/6 | 2.1% |
| EOG | **1/2** | **3.0%** | 0/2 | 0.0% | 0/2 | 0.0% | 0/2 | 0.0% | 0/2 | 0.0% | 0/2 | 0.0% | 0/2 | 0.0% | 0/2 | 0.0% | 0/2 | 0.0% |
| Motion | **11/14** | 2.4% | 4/14 | 3.1% | 5/14 | 4.6% | 3/14 | 0.8% | 3/14 | 3.8% | 4/14 | 2.1% | 4/14 | 2.6% | 5/14 | 2.8% | 6/14 | 2.5% |
| EPG | 1/2 | 5.2% | 1/2 | 1.2% | **2/2** | **7.6%** | 1/2 | 1.2% | 0/2 | 0.0% | 0/2 | 0.0% | 1/2 | 2.4% | 0/2 | 0.0% | 0/2 | 0.0% |
| Power | **0/1** | **0.0%** | **0/1** | **0.0%** | **0/1** | **0.0%** | **0/1** | **0.0%** | **0/1** | **0.0%** | **0/1** | **0.0%** | **0/1** | **0.0%** | **0/1** | **0.0%** | **0/1** | **0.0%** |
| Spectrum | **3/4** | 4.4% | 1/4 | 4.0% | 2/4 | 5.4% | 1/4 | 2.8% | 1/4 | 0.8% | 2/4 | 2.1% | 1/4 | 12.0% | 1/4 | 0.3% | 1/4 | 2.0% |

Table 15: Mean worsened accuracy change across dataset domains (Moment). Smallest reductions in bold.

| Type | ASCENSION | | Diffusion-TS | | FAA | | ImagenTime | | KoVAE | | LA | | Time-DDPM | | TTS-GAN | | VaDE | |
|---|---|---|---|---|---|---|---|---|---|---|---|---|---|---|---|---|---|---|
| | ↓Nb | ↓ $\overline{\Delta Acc}$ | ↓Nb | ↓ $\overline{\Delta Acc}$ | ↓Nb | ↓ $\overline{\Delta Acc}$ | ↓Nb | ↓ $\overline{\Delta Acc}$ | ↓Nb | ↓ $\overline{\Delta Acc}$ | ↓Nb | ↓ $\overline{\Delta Acc}$ | ↓Nb | ↓ $\overline{\Delta Acc}$ | ↓Nb | ↓ $\overline{\Delta Acc}$ | ↓Nb | ↓ $\overline{\Delta Acc}$ |
| Device | **1/8** | **-0.5%** | 5/8 | -1.6% | 3/8 | -4.5% | 7/8 | -2.4% | 7/8 | -3.4% | 5/8 | -2.3% | 6/8 | -1.9% | 5/8 | -2.5% | 6/8 | -1.7% |
| Image | **5/30** | **-1.1%** | 18/30 | -11.7% | 12/30 | -9.8% | 19/30 | -6.0% | 21/30 | -10.9% | 20/30 | -4.9% | 17/30 | -5.6% | 16/30 | -4.2% | 17/30 | -3.7% |
| Simulated | **2/8** | **-0.8%** | 6/8 | -15.3% | 4/8 | -12.4% | 5/8 | -9.8% | 6/8 | -12.6% | 6/8 | -9.4% | 5/8 | -8.5% | 4/8 | -12.4% | 4/8 | -9.9% |
| Spectro | **2/8** | **-7.2%** | 6/8 | -15.4% | 4/8 | -24.2% | 7/8 | -13.8% | 7/8 | -9.5% | 6/8 | -10.2% | 7/8 | -8.1% | 6/8 | -8.7% | 6/8 | -5.9% |
| Sensor | **4/19** | **-0.8%** | 13/19 | -6.9% | 7/19 | -9.7% | 12/19 | -5.5% | 7/19 | -5.0% | 9/19 | -4.1% | 9/19 | -5.4% | 8/19 | -3.3% | 9/19 | -4.0% |
| ECG | **1/6** | **-0.1%** | 5/6 | -7.8% | 3/6 | -4.2% | 4/6 | -4.6% | 4/6 | -31.2% | 4/6 | -4.2% | 2/6 | -59.1% | **1/6** | -1.6% | 4/6 | -20.9% |
| EOG | **1/2** | **-1.1%** | 2/2 | -10.2% | 2/2 | -12.6% | 2/2 | -9.0% | 2/2 | -9.4% | 2/2 | -10.4% | 2/2 | -7.3% | 2/2 | -8.8% | 2/2 | -7.5% |
| Motion | **2/14** | **-0.9%** | 9/14 | -3.8% | 8/14 | -15.9% | 10/14 | -4.2% | 10/14 | -25.2% | 8/14 | -5.2% | 9/14 | -2.6% | 7/14 | -4.1% | 5/14 | -3.8% |
| EPG | **0/2** | **0.0%** | 1/2 | -4.8% | **0/2** | **0.0%** | 1/2 | -15.3% | 2/2 | -5.2% | 2/2 | -3.8% | 1/2 | -12.4% | 1/2 | -3.6% | 2/2 | -8.4% |
| Power | **1/1** | **-2.8%** | 1/1 | -3.3% | 1/1 | -4.4% | **1/1** | **-2.8%** | 1/1 | -4.4% | 1/1 | -3.9% | 1/1 | -4.4% | 1/1 | -6.1% | 1/1 | -6.1% |
| Spectrum | **1/4** | **-0.2%** | 3/4 | -3.9% | 2/4 | -14.9% | 2/4 | -1.6% | 3/4 | -5.0% | 2/4 | -1.7% | 3/4 | -1.6% | 3/4 | -4.1% | 3/4 | -1.1% |

Table 16: Macro-domain breakdown – ResNet: number of improved datasets (↑Nb) out of total per macro-domain, and net mean accuracy change ($\overline{\Delta\text{Acc}}$). Macro-domains group the 11 UCR raw types as follows: Industrial = Sensor + Device + Simulated + Power; Spectral = Spectro + Spectrum; Biomedical = ECG + EPG + EOG; Image and Motion are unchanged.

| Macro | ASCENSION | | Diffusion-TS | | FAA | | ImagenTime | | KoVAE | | LA | | Time-DDPM | | TTS-GAN | | VaDE | |
|---|---|---|---|---|---|---|---|---|---|---|---|---|---|---|---|---|---|---|
| | ↑Nb | $\overline{\Delta\text{Acc}}$ | ↑Nb | $\overline{\Delta\text{Acc}}$ | ↑Nb | $\overline{\Delta\text{Acc}}$ | ↑Nb | $\overline{\Delta\text{Acc}}$ | ↑Nb | $\overline{\Delta\text{Acc}}$ | ↑Nb | $\overline{\Delta\text{Acc}}$ | ↑Nb | $\overline{\Delta\text{Acc}}$ | ↑Nb | $\overline{\Delta\text{Acc}}$ | ↑Nb | $\overline{\Delta\text{Acc}}$ |
| Industrial | 17/36 | 0.7% | 10/36 | -4.8% | 8/36 | -2.6% | 6/36 | -3.6% | 15/36 | 0.2% | 8/36 | -1.2% | 9/36 | -11.1% | 11/36 | -2.2% | 14/36 | -1.9% |
| Image | 22/30 | 2.4% | 8/30 | -8.3% | 4/30 | -4.3% | 5/30 | -4.8% | 11/30 | -0.3% | 3/30 | -3.1% | 11/30 | -9.5% | 10/30 | -2.8% | 17/30 | -3.4% |
| Motion | 8/14 | 0.8% | 0/14 | -4.9% | 1/14 | -7.6% | 0/14 | -3.3% | 5/14 | -0.3% | 0/14 | -3.9% | 7/14 | -8.3% | 1/14 | -13.3% | 5/14 | -0.5% |
| Spectral | 7/12 | 7.1% | 2/12 | -5.8% | 5/12 | -7.7% | 3/12 | -3.3% | 4/12 | -2.1% | 1/12 | -2.3% | 7/12 | 4.4% | 3/12 | -4.0% | 6/12 | 1.7% |
| Biomedical | 5/10 | -0.1% | 1/10 | -11.5% | 1/10 | -11.7% | 1/10 | -7.3% | 3/10 | -0.6% | 2/10 | -3.2% | 4/10 | -6.3% | 1/10 | -8.2% | 2/10 | -6.7% |

Table 17: Macro-domain breakdown – FCN: number of improved datasets (↑Nb) out of total per macro-domain, and net mean accuracy change ($\overline{\Delta\text{Acc}}$). Macro-domains group the 11 UCR raw types as follows: Industrial = Sensor + Device + Simulated + Power; Spectral = Spectro + Spectrum; Biomedical = ECG + EPG + EOG; Image and Motion are unchanged.

| Macro | ASCENSION | | Diffusion-TS | | FAA | | ImagenTime | | KoVAE | | LA | | Time-DDPM | | TTS-GAN | | VaDE | |
|---|---|---|---|---|---|---|---|---|---|---|---|---|---|---|---|---|---|---|
| | ↑Nb | $\overline{\Delta\text{Acc}}$ | ↑Nb | $\overline{\Delta\text{Acc}}$ | ↑Nb | $\overline{\Delta\text{Acc}}$ | ↑Nb | $\overline{\Delta\text{Acc}}$ | ↑Nb | $\overline{\Delta\text{Acc}}$ | ↑Nb | $\overline{\Delta\text{Acc}}$ | ↑Nb | $\overline{\Delta\text{Acc}}$ | ↑Nb | $\overline{\Delta\text{Acc}}$ | ↑Nb | $\overline{\Delta\text{Acc}}$ |
| Industrial | 16/36 | 0.1% | 15/36 | -3.4% | 7/36 | -7.7% | 14/36 | -0.4% | 13/36 | -0.4% | 13/36 | -0.5% | 9/36 | -14.6% | 14/36 | -1.2% | 12/36 | -1.9% |
| Image | 17/30 | 0.5% | 13/30 | -11.2% | 2/30 | -14.3% | 9/30 | -1.2% | 12/30 | 0.0% | 14/30 | 0.2% | 13/30 | -4.0% | 11/30 | -1.7% | 15/30 | -2.7% |
| Motion | 7/14 | -0.3% | 1/14 | -5.5% | 2/14 | -9.5% | 6/14 | -0.8% | 6/14 | 0.2% | 9/14 | 1.5% | 9/14 | -0.7% | 3/14 | -7.9% | 9/14 | 1.0% |
| Spectral | 10/12 | 8.9% | 6/12 | -2.5% | 3/12 | -14.3% | 6/12 | 0.6% | 6/12 | 1.0% | 3/12 | -1.6% | 8/12 | 3.0% | 4/12 | -3.5% | 5/12 | 1.0% |
| Biomedical | 6/10 | 1.6% | 1/10 | -11.1% | 0/10 | -16.4% | 3/10 | 0.2% | 4/10 | -1.1% | 5/10 | 1.1% | 3/10 | -2.6% | 4/10 | -4.9% | 1/10 | -5.2% |

Table 18: Macro-domain breakdown – Moment: number of improved datasets (↑Nb) out of total per macro-domain, and net mean accuracy change ($\overline{\Delta\text{Acc}}$). Macro-domains group the 11 UCR raw types as follows: Industrial = Sensor + Device + Simulated + Power; Spectral = Spectro + Spectrum; Biomedical = ECG + EPG + EOG; Image and Motion are unchanged.

| Macro | ASCENSION | | Diffusion-TS | | FAA | | ImagenTime | | KoVAE | | LA | | Time-DDPM | | TTS-GAN | | VaDE | |
|---|---|---|---|---|---|---|---|---|---|---|---|---|---|---|---|---|---|---|
| | ↑Nb | $\overline{\Delta\text{Acc}}$ | ↑Nb | $\overline{\Delta\text{Acc}}$ | ↑Nb | $\overline{\Delta\text{Acc}}$ | ↑Nb | $\overline{\Delta\text{Acc}}$ | ↑Nb | $\overline{\Delta\text{Acc}}$ | ↑Nb | $\overline{\Delta\text{Acc}}$ | ↑Nb | $\overline{\Delta\text{Acc}}$ | ↑Nb | $\overline{\Delta\text{Acc}}$ | ↑Nb | $\overline{\Delta\text{Acc}}$ |
| Industrial | 23/36 | 2.8% | 9/36 | -4.8% | 18/36 | 1.7% | 9/36 | -3.3% | 12/36 | -2.7% | 13/36 | -2.1% | 13/36 | -1.4% | 16/36 | -1.1% | 14/36 | -1.6% |
| Image | 24/30 | 2.6% | 11/30 | -5.4% | 16/30 | -0.8% | 9/30 | -3.0% | 9/30 | -6.2% | 10/30 | -2.0% | 12/30 | -1.6% | 14/30 | -0.4% | 12/30 | -0.6% |
| Motion | 11/14 | 1.8% | 4/14 | -1.6% | 5/14 | -7.4% | 3/14 | -2.8% | 3/14 | -17.2% | 4/14 | -2.4% | 4/14 | -0.9% | 5/14 | -1.1% | 6/14 | -0.3% |
| Spectral | 8/12 | 2.0% | 2/12 | -8.3% | 6/12 | -7.3% | 1/12 | -8.1% | 1/12 | -6.8% | 2/12 | -5.0% | 1/12 | -4.1% | 2/12 | -5.1% | 1/12 | -3.1% |
| Biomedical | 6/10 | 1.8% | 2/10 | -6.3% | 5/10 | 0.0% | 3/10 | -4.4% | 2/10 | -15.2% | 2/10 | -4.4% | 5/10 | -13.7% | 4/10 | -1.1% | 2/10 | -11.1% |

Table 19: UCR dataset domains and their selected representative datasets

| Type | Representative dataset | Description |
|---|---|---|
| Device | ACSF1 | Measurements of alternating current signals for predictive maintenance |
| Image | BeetleFly | Shape-based image classification of beetle and fly outlines |
| Simulated | UMD | Simulated control processes data |
| Spectro | Ham | Spectroscopy data to identify types of ham based on chemical properties |
| Sensor | Car | Sensor readings collected from a car, used for detecting driving conditions |
| ECG | ECG200 | Electrocardiogram (ECG) readings used to detect heart abnormalities |
| EOG | EOGVerticalSignal | Electrooculography (EOG) signals capturing eye movement patterns |
| Motion | Worms | Motion sensor data capturing worm movements for classification |
| EPG | InsectEPGRegularTrain | Electrical penetration graph (EPG) signals capturing insect feeding behaviour |
| Power | PowerCons | Power consumption measurements for energy usage |
| Spectrum | SemgHandMovementCh2 | Electromyography (EMG) data of hand movements, recorded across channels |

Tables 20, 21, and 22 report the best $(\text{iter.}, \alpha)$ configuration per dataset for each of the three classifiers used in Table 1, together with the corresponding accuracy improvement $(\overline{\Delta \text{Acc}} = \text{aug\_acc} - \text{base\_acc})$.

**Augmentation budget.** All methods are configured with the same per-class generation budget: at each outer iteration, $10 \cdot n/Y$ synthetic samples are produced per class (where $n$ is the train-set size and $Y$ the number of classes). For ASCENSION, the Bayes-region rejection step (Definition 2) discards out-of-region candidates, so the realised number of synthetic samples that ASCENSION adds to the training set is in practice smaller than this upper bound and varies across datasets (median acceptance rate $\approx 0.65\text{-}0.70$, i.e. roughly $7 \cdot n/Y$ surviving samples per class; see Appendix G).

Table 20: Best (iter., $\alpha$) configuration per dataset for the ResNet classifier in Table 1, with the corresponding accuracy improvement ($\overline{\Delta\text{Acc}}$ = augmentation_accuracy − base_accuracy).

| # | Dataset | (iter., $\alpha$) | $\overline{\Delta\text{Acc}}$ | # | Dataset | (iter., $\alpha$) | $\overline{\Delta\text{Acc}}$ |
|---|---|---|---|---|---|---|---|
| D1 | ACSF1 | (2, 3.2) | 0.03 | D52 | Meat | (1, 1.5) | -0.00 |
| D2 | Adiac | (1, 2) | 0.02 | D53 | MedicalImages | (1, 4.2) | 0.02 |
| D3 | ArrowHead | (3, 4.7) | 0.02 | D54 | MiddlePhalanxOutlineAgeGroup | (2, 4.9) | 0.01 |
| D4 | BME | (3, 1.2) | -0.01 | D55 | MiddlePhalanxOutlineCorrect | (1, 3.7) | -0.01 |
| D5 | Beef | (1, 1.3) | 0.20 | D56 | MiddlePhalanxTW | (1, 4) | 0.01 |
| D6 | BeetleFly | (1, 1.2) | 0.00 | D57 | MixedShapesRegularTrain | (1, 1.3) | 0.01 |
| D7 | BirdChicken | (2, 3.4) | 0.25 | D58 | MixedShapesSmallTrain | (1, 1.5) | 0.02 |
| D8 | Car | (1, 1.2) | 0.07 | D59 | MoteStrain | (2, 1.8) | 0.00 |
| D9 | CBF | (1, 1.1) | -0.00 | D60 | NonInvasiveFetalECGThorax1 | (1, 1.5) | 0.03 |
| D10 | ChlorineConcentration | (3, 2.2) | 0.03 | D61 | NonInvasiveFetalECGThorax2 | (1, 4.4) | 0.01 |
| D11 | CinCECGTorso | (3, 4) | 0.01 | D62 | OSULeaf | (1, 1) | -0.02 |
| D12 | Coffee | (2, 4.3) | 0.00 | D63 | OliveOil | (2, 1.7) | 0.07 |
| D13 | Computers | (1, 4.4) | 0.02 | D64 | PhalangesOutlinesCorrect | (2, 4.5) | -0.01 |
| D14 | Crop | (3, 3.4) | 0.01 | D65 | Plane | (1, 4.1) | 0.00 |
| D15 | DistalPhalanxOutlineAgeGroup | (2, 3.9) | -0.01 | D66 | PowerCons | (1, 3.9) | 0.02 |
| D16 | DistalPhalanxOutlineCorrect | (1, 3.3) | 0.02 | D67 | ProximalPhalanxOutlineAgeGroup | (2, 3.7) | 0.01 |
| D17 | DistalPhalanxTW | (1, 2.9) | 0.01 | D68 | ProximalPhalanxOutlineCorrect | (3, 3.6) | -0.01 |
| D18 | ECG200 | (1, 1.1) | 0.00 | D69 | ProximalPhalanxTW | (1, 4.6) | 0.02 |
| D19 | ECG5000 | (1, 2.5) | 0.00 | D70 | RefrigerationDevices | (1, 2.8) | -0.02 |
| D20 | ECGFiveDays | (2, 4.6) | 0.08 | D71 | Rock | (2, 3) | 0.06 |
| D21 | EOGHorizontalSignal | (2, 4.7) | -0.11 | D72 | ScreenType | (1, 2.4) | -0.02 |
| D22 | EOGVerticalSignal | (2, 4.3) | -0.03 | D73 | SemgHandGenderCh2 | (1, 1.5) | 0.03 |
| D23 | Earthquakes | (3, 3.6) | 0.03 | D74 | SemgHandMovementCh2 | (1, 4.2) | -0.02 |
| D24 | ElectricDevices | (3, 2.9) | -0.02 | D75 | SemgHandSubjectCh2 | (2, 1.4) | -0.04 |
| D25 | EthanolLevel | (2, 4.2) | 0.38 | D76 | ShapeletSim | (1, 2.8) | -0.01 |
| D26 | FaceAll | (1, 2) | 0.09 | D77 | ShapesAll | (1, 3.3) | 0.02 |
| D27 | FaceFour | (1, 2.9) | 0.09 | D78 | SmallKitchenAppliances | (2, 4.7) | 0.02 |
| D28 | FacesUCR | (1, 3.7) | -0.01 | D79 | SmoothSubspace | (2, 1.3) | -0.01 |
| D29 | Fish | (1, 3.7) | 0.10 | D80 | SonyAIBORobotSurface1 | (1, 3.4) | 0.00 |
| D30 | FordA | (1, 1.5) | -0.00 | D81 | SonyAIBORobotSurface2 | (3, 1.3) | -0.00 |
| D31 | FordB | (1, 1.5) | 0.02 | D82 | StarlightCurves | (1, 1.5) | -0.01 |
| D32 | FreezerRegularTrain | (2, 1.6) | 0.00 | D83 | Strawberry | (1, 3.9) | 0.00 |
| D33 | FreezerSmallTrain | (1, 2.8) | -0.04 | D84 | SwedishLeaf | (1, 1.9) | 0.00 |
| D34 | GunPoint | (1, 3.6) | 0.01 | D85 | Symbols | (2, 2.2) | 0.02 |
| D35 | GunPointAgeSpan | (1, 4.4) | -0.00 | D86 | SyntheticControl | (1, 2.7) | -0.00 |
| D36 | GunPointMaleVersusFemale | (1, 2.1) | 0.00 | D87 | ToeSegmentation1 | (1, 2.5) | -0.01 |
| D37 | GunPointOldVersusYoung | (1, 3.2) | 0.00 | D88 | ToeSegmentation2 | (2, 2.5) | 0.00 |
| D38 | Ham | (3, 1.7) | 0.05 | D89 | Trace | (1, 2.7) | 0.00 |
| D39 | HandOutlines | (1, 1.6) | 0.00 | D90 | TwoLeadECG | (1, 1.1) | 0.06 |
| D40 | Haptics | (1, 1.6) | 0.03 | D91 | TwoPatterns | (1, 3.8) | 0.01 |
| D41 | Herring | (1, 2.2) | 0.05 | D92 | UMD | (2, 3.4) | 0.00 |
| D42 | HouseTwenty | (2, 3.5) | 0.03 | D93 | UWaveGestureLibraryAll | (1, 3.9) | 0.02 |
| D43 | InlineSkate | (1, 1.5) | -0.02 | D94 | UWaveGestureLibraryX | (1, 3) | 0.02 |
| D44 | InsectWingbeatSound | (3, 2) | 0.00 | D95 | UWaveGestureLibraryY | (1, 1.9) | 0.01 |
| D45 | InsectEPGRegularTrain | (1, 1) | 0.00 | D96 | UWaveGestureLibraryZ | (1, 4.5) | 0.01 |
| D46 | InsectEPGSmallTrain | (1, 2.7) | 0.00 | D97 | Wafer | (1, 1.5) | 0.00 |
| D47 | ItalyPowerDemand | (1, 3) | -0.00 | D98 | Wine | (1, 2.1) | 0.13 |
| D48 | LargeKitchenAppliances | (2, 5) | -0.01 | D99 | WordSynonyms | (1, 2.9) | -0.03 |
| D49 | Lightning2 | (2, 3.7) | 0.05 | D100 | Worms | (1, 4.7) | 0.04 |
| D50 | Lightning7 | (2, 4.9) | -0.01 | D101 | WormsTwoClass | (2, 3.7) | 0.00 |
| D51 | Mallat | (1, 4.7) | -0.00 | D102 | Yoga | (1, 1.7) | 0.02 |

Table 21: Best (iter., $\alpha$) configuration per dataset for the FCN classifier in Table 1, with the corresponding accuracy improvement ($\overline{\Delta\text{Acc}}$ = augmentation_accuracy − base_accuracy).

| # | Dataset | (iter., $\alpha$) | $\overline{\Delta\text{Acc}}$ | # | Dataset | (iter., $\alpha$) | $\overline{\Delta\text{Acc}}$ |
|---|---|---|---|---|---|---|---|
| D1 | ACSF1 | (2, 1.5) | 0.02 | D52 | Meat | (1, 1.7) | 0.02 |
| D2 | Adiac | (1, 2.6) | 0.04 | D53 | MedicalImages | (1, 1.5) | -0.03 |
| D3 | ArrowHead | (3, 1.4) | -0.02 | D54 | MiddlePhalanxOutlineAgeGroup | (1, 2) | -0.01 |
| D4 | BME | (2, 1.7) | 0.01 | D55 | MiddlePhalanxOutlineCorrect | (1, 1.8) | 0.02 |
| D5 | Beef | (1, 2.9) | 0.23 | D56 | MiddlePhalanxTW | (1, 1.7) | 0.03 |
| D6 | BeetleFly | (1, 1.7) | 0.05 | D57 | MixedShapesRegularTrain | (1, 1.5) | 0.01 |
| D7 | BirdChicken | (3, 2.3) | 0.20 | D58 | MixedShapesSmallTrain | (1, 1.3) | -0.05 |
| D8 | Car | (1, 1) | -0.02 | D59 | MoteStrain | (1, 2.4) | 0.01 |
| D9 | CBF | (1, 1.1) | 0.00 | D60 | NonInvasiveFetalECGThorax1 | (1, 1.2) | 0.06 |
| D10 | ChlorineConcentration | (1, 1.3) | -0.02 | D61 | NonInvasiveFetalECGThorax2 | (2, 1.5) | 0.07 |
| D11 | CinCECGTorso | (2, 1) | 0.03 | D62 | OSULeaf | (1, 1.8) | -0.16 |
| D12 | Coffee | (1, 2.1) | 0.07 | D63 | OliveOil | (2, 1.6) | 0.07 |
| D13 | Computers | (1, 1.5) | 0.09 | D64 | PhalangesOutlinesCorrect | (2, 2) | 0.01 |
| D14 | Crop | (2, 1.3) | 0.01 | D65 | Plane | (1, 2.8) | 0.00 |
| D15 | DistalPhalanxOutlineAgeGroup | (1, 1.1) | 0.01 | D66 | PowerCons | (1, 2.2) | -0.00 |
| D16 | DistalPhalanxOutlineCorrect | (2, 1.2) | -0.01 | D67 | ProximalPhalanxOutlineAgeGroup | (1, 1.4) | 0.01 |
| D17 | DistalPhalanxTW | (3, 1.2) | -0.01 | D68 | ProximalPhalanxOutlineCorrect | (1, 1.8) | 0.01 |
| D18 | ECG200 | (3, 1.7) | -0.02 | D69 | ProximalPhalanxTW | (1, 4.7) | 0.00 |
| D19 | ECG5000 | (1, 2.5) | 0.00 | D70 | RefrigerationDevices | (2, 1.1) | 0.03 |
| D20 | ECGFiveDays | (3, 4) | 0.02 | D71 | Rock | (1, 3.2) | 0.04 |
| D21 | EOGHorizontalSignal | (1, 1.4) | -0.01 | D72 | ScreenType | (1, 2.3) | -0.01 |
| D22 | EOGVerticalSignal | (2, 1.2) | 0.02 | D73 | SemgHandGenderCh2 | (2, 1.5) | 0.07 |
| D23 | Earthquakes | (1, 2.2) | -0.01 | D74 | SemgHandMovementCh2 | (1, 1.4) | -0.02 |
| D24 | ElectricDevices | (1, 1.5) | 0.02 | D75 | SemgHandSubjectCh2 | (1, 1.6) | 0.00 |
| D25 | EthanolLevel | (1, 4.8) | 0.43 | D76 | ShapeletSim | (1, 2.1) | 0.00 |
| D26 | FaceAll | (1, 1.2) | 0.00 | D77 | ShapesAll | (3, 2.9) | 0.05 |
| D27 | FaceFour | (1, 1.4) | -0.01 | D78 | SmallKitchenAppliances | (2, 1.3) | -0.00 |
| D28 | FacesUCR | (1, 1.9) | 0.01 | D79 | SmoothSubspace | (2, 2.4) | 0.00 |
| D29 | Fish | (2, 1.2) | -0.03 | D80 | SonyAIBORobotSurface1 | (1, 2.2) | 0.02 |
| D30 | FordA | (1, 1.5) | -0.00 | D81 | SonyAIBORobotSurface2 | (2, 2.4) | 0.02 |
| D31 | FordB | (1, 1.2) | -0.00 | D82 | StarlightCurves | (2, 1.3) | -0.02 |
| D32 | FreezerRegularTrain | (1, 1.4) | 0.00 | D83 | Strawberry | (2, 4.9) | 0.01 |
| D33 | FreezerSmallTrain | (1, 1) | -0.03 | D84 | SwedishLeaf | (1, 1.5) | -0.05 |
| D34 | GunPoint | (2, 1.7) | 0.01 | D85 | Symbols | (1, 1.1) | 0.00 |
| D35 | GunPointAgeSpan | (1, 3.4) | 0.00 | D86 | SyntheticControl | (2, 1.8) | 0.00 |
| D36 | GunPointMaleVersusFemale | (2, 1) | 0.01 | D87 | ToeSegmentation1 | (1, 1.1) | 0.00 |
| D37 | GunPointOldVersusYoung | (1, 4.4) | 0.00 | D88 | ToeSegmentation2 | (2, 4.6) | 0.03 |
| D38 | Ham | (1, 3) | 0.03 | D89 | Trace | (2, 1.9) | 0.00 |
| D39 | HandOutlines | (1, 1) | 0.01 | D90 | TwoLeadECG | (1, 2.3) | 0.03 |
| D40 | Haptics | (3, 1.4) | -0.01 | D91 | TwoPatterns | (1, 1.5) | 0.00 |
| D41 | Herring | (1, 1.2) | 0.06 | D92 | UMD | (1, 2.8) | -0.01 |
| D42 | HouseTwenty | (1, 1.6) | 0.01 | D93 | UWaveGestureLibraryAll | (1, 1.6) | -0.10 |
| D43 | InlineSkate | (1, 1.2) | -0.01 | D94 | UWaveGestureLibraryX | (2, 1.1) | 0.01 |
| D44 | InsectWingbeatSound | (1, 1.1) | -0.03 | D95 | UWaveGestureLibraryY | (2, 1.3) | -0.04 |
| D45 | InsectEPGRegularTrain | (1, 1.3) | 0.00 | D96 | UWaveGestureLibraryZ | (1, 1.6) | -0.04 |
| D46 | InsectEPGSmallTrain | (1, 1.1) | 0.00 | D97 | Wafer | (1, 3.6) | -0.00 |
| D47 | ItalyPowerDemand | (3, 2.8) | 0.00 | D98 | Wine | (3, 3.5) | 0.11 |
| D48 | LargeKitchenAppliances | (1, 3) | -0.04 | D99 | WordSynonyms | (3, 2.7) | -0.00 |
| D49 | Lightning2 | (1, 1.8) | -0.03 | D100 | Worms | (1, 1.1) | 0.06 |
| D50 | Lightning7 | (1, 1.1) | 0.03 | D101 | WormsTwoClass | (3, 2.2) | 0.01 |
| D51 | Mallat | (1, 2.3) | 0.00 | D102 | Yoga | (1, 2.8) | -0.01 |

Table 22: Best (iter., $\alpha$) configuration per dataset for the Moment classifier in Table 1, with the corresponding accuracy improvement ($\overline{\Delta\text{Acc}} = \text{augmentation\_accuracy} - \text{base\_accuracy}$).

| # | Dataset | (iter., $\alpha$) | $\overline{\Delta\text{Acc}}$ | # | Dataset | (iter., $\alpha$) | $\overline{\Delta\text{Acc}}$ |
|---|---|---|---|---|---|---|---|
| D1 | ACSF1 | (1, 3.8) | 0.00 | D52 | Meat | (2, 3.5) | 0.05 |
| D2 | Adiac | (1, 4.4) | 0.01 | D53 | MedicalImages | (1, 3.5) | 0.02 |
| D3 | ArrowHead | (1, 1.9) | 0.01 | D54 | MiddlePhalanxOutlineAgeGroup | (1, 2.5) | 0.03 |
| D4 | BME | (1, 1.9) | 0.08 | D55 | MiddlePhalanxOutlineCorrect | (2, 3.7) | 0.00 |
| D5 | Beef | (1, 2.7) | 0.07 | D56 | MiddlePhalanxTW | (1, 3.2) | 0.04 |
| D6 | BeetleFly | (3, 2.3) | 0.15 | D57 | MixedShapesRegularTrain | (1, 1.9) | 0.00 |
| D7 | BirdChicken | (2, 1.7) | 0.10 | D58 | MixedShapesSmallTrain | (2, 1.9) | 0.00 |
| D8 | Car | (1, 3.6) | -0.02 | D59 | MoteStrain | (2, 1.6) | 0.06 |
| D9 | CBF | (1, 1.3) | -0.00 | D60 | NonInvasiveFetalECGThorax1 | (1, 2.3) | 0.00 |
| D10 | ChlorineConcentration | (1, 4.3) | 0.06 | D61 | NonInvasiveFetalECGThorax2 | (1, 2.7) | 0.00 |
| D11 | CinCECGTorso | (1, 3.7) | 0.00 | D62 | OSULeaf | (2, 2.9) | 0.05 |
| D12 | Coffee | (1, 4.5) | 0.04 | D63 | OliveOil | (2, 2.9) | 0.00 |
| D13 | Computers | (1, 2.5) | 0.03 | D64 | PhalangesOutlinesCorrect | (1, 3.2) | 0.03 |
| D14 | Crop | (1, 4.4) | -0.00 | D65 | Plane | (1, 3) | 0.01 |
| D15 | DistalPhalanxOutlineAgeGroup | (1, 2.7) | 0.06 | D66 | PowerCons | (2, 1.9) | -0.03 |
| D16 | DistalPhalanxOutlineCorrect | (3, 1.5) | 0.03 | D67 | ProximalPhalanxOutlineAgeGroup | (2, 1.7) | 0.05 |
| D17 | DistalPhalanxTW | (1, 4) | -0.01 | D68 | ProximalPhalanxOutlineCorrect | (1, 4.5) | 0.01 |
| D18 | ECG200 | (1, 1.7) | 0.02 | D69 | ProximalPhalanxTW | (1, 3.2) | 0.02 |
| D19 | ECG5000 | (1, 4.6) | -0.00 | D70 | RefrigerationDevices | (1, 3.9) | -0.01 |
| D20 | ECGFiveDays | (1, 2.9) | 0.08 | D71 | Rock | (1, 4.4) | 0.12 |
| D21 | EOGHorizontalSignal | (1, 2) | -0.01 | D72 | ScreenType | (1, 2.8) | 0.00 |
| D22 | EOGVerticalSignal | (2, 2.9) | 0.03 | D73 | SemgHandGenderCh2 | (2, 1.2) | 0.01 |
| D23 | Earthquakes | (3, 2.2) | 0.03 | D74 | SemgHandMovementCh2 | (1, 3.4) | -0.00 |
| D24 | ElectricDevices | (1, 4.6) | 0.01 | D75 | SemgHandSubjectCh2 | (1, 1.3) | 0.01 |
| D25 | EthanolLevel | (2, 3.2) | 0.01 | D76 | ShapeletSim | (1, 2.8) | -0.01 |
| D26 | FaceAll | (1, 2.2) | 0.01 | D77 | ShapesAll | (1, 1.6) | 0.01 |
| D27 | FaceFour | (1, 3.6) | 0.05 | D78 | SmallKitchenAppliances | (1, 3.5) | 0.02 |
| D28 | FacesUCR | (1, 2.6) | -0.01 | D79 | SmoothSubspace | (1, 3) | 0.01 |
| D29 | Fish | (1, 2.8) | -0.01 | D80 | SonyAIBORobotSurface1 | (3, 2.3) | 0.27 |
| D30 | FordA | (1, 1.5) | 0.00 | D81 | SonyAIBORobotSurface2 | (2, 4.3) | 0.08 |
| D31 | FordB | (1, 4) | -0.00 | D82 | StarlightCurves | (1, 4.3) | 0.01 |
| D32 | FreezerRegularTrain | (1, 3.2) | 0.00 | D83 | Strawberry | (1, 5) | -0.03 |
| D33 | FreezerSmallTrain | (2, 4.3) | 0.05 | D84 | SwedishLeaf | (1, 4.4) | 0.01 |
| D34 | GunPoint | (1, 2.6) | 0.03 | D85 | Symbols | (1, 4.4) | 0.01 |
| D35 | GunPointAgeSpan | (1, 4.4) | 0.01 | D86 | SyntheticControl | (1, 4.4) | 0.00 |
| D36 | GunPointMaleVersusFemale | (2, 4.8) | 0.00 | D87 | ToeSegmentation1 | (1, 1.3) | 0.01 |
| D37 | GunPointOldVersusYoung | (1, 3.6) | 0.00 | D88 | ToeSegmentation2 | (1, 3.5) | 0.04 |
| D38 | Ham | (1, 1.2) | 0.09 | D89 | Trace | (1, 4.5) | -0.01 |
| D39 | HandOutlines | (1, 2.9) | 0.00 | D90 | TwoLeadECG | (2, 1.2) | 0.13 |
| D40 | Haptics | (2, 3.4) | 0.09 | D91 | TwoPatterns | (1, 2.6) | 0.00 |
| D41 | Herring | (1, 4.8) | 0.14 | D92 | UMD | (1, 3.6) | 0.00 |
| D42 | HouseTwenty | (1, 1.5) | 0.00 | D93 | UWaveGestureLibraryAll | (1, 1.2) | -0.00 |
| D43 | InlineSkate | (1, 2.1) | 0.05 | D94 | UWaveGestureLibraryX | (1, 2.4) | 0.00 |
| D44 | InsectWingbeatSound | (1, 3.5) | 0.01 | D95 | UWaveGestureLibraryY | (1, 3.2) | 0.00 |
| D45 | InsectEPGRegularTrain | (1, 2.5) | 0.00 | D96 | UWaveGestureLibraryZ | (1, 1.5) | 0.00 |
| D46 | InsectEPGSmallTrain | (1, 2) | 0.05 | D97 | Wafer | (1, 4.1) | -0.00 |
| D47 | ItalyPowerDemand | (3, 2.4) | 0.03 | D98 | Wine | (3, 1.5) | -0.11 |
| D48 | LargeKitchenAppliances | (1, 2.1) | 0.02 | D99 | WordSynonyms | (1, 3) | 0.01 |
| D49 | Lightning2 | (1, 4.9) | 0.07 | D100 | Worms | (1, 3.6) | 0.03 |
| D50 | Lightning7 | (1, 2.9) | 0.04 | D101 | WormsTwoClass | (1, 2.5) | -0.01 |
| D51 | Mallat | (1, 4.7) | 0.06 | D102 | Yoga | (1, 1.1) | -0.03 |

# K    Extended hyperparameter sensitivity analysis

### K.1    $\alpha$-scaling mechanism ablation study

Figures 12 and 13 show 3D plots of classifiers' performance as a function of $\alpha$ and iteration count for ASCENSION$_{\text{Emb}}$, ASCENSION$_{\text{FCN-Emb}}$, and ASCENSION$_{\text{ResNet-Emb}}$. It appears that the relationship between $\alpha$, iteration count, and performance improvement is not trivial to generalise, although it remains relatively stable. Increasing $\alpha$ can facilitate more thorough boundary exploration, but excessive values may sometimes degrade performance, as evidenced e.g. in Figure 12 (see ASCENSION$_{\text{ResNet-Emb}}$) and Figure 13 (see ASCENSION$_{\text{Emb}}$ and ASCENSION$_{\text{ResNet-Emb}}$).

Using the range of $\alpha \in \{1, 2, 3, 4, 5\}$ and iteration counts $\in \{1, 2, \ldots, 9\}$, the (iteration, $\alpha$) pair that yields the highest accuracy improvement for each dataset using the ASCENSION$_{\text{ResNet-Emb}}$ classifier is reported in Table 23 – see column (iter., $\alpha$). The corresponding accuracy gain from using the $\alpha$-scaling mechanism, compared to not using it, is shown in column "Gain" and computed as the difference between the best and the worst accuracy over the steps.

ASCENSION$_{\text{Emb}}$.         ASCENSION$_{\text{FCN-Emb}}$.         ASCENSION$_{\text{ResNet-Emb}}$.

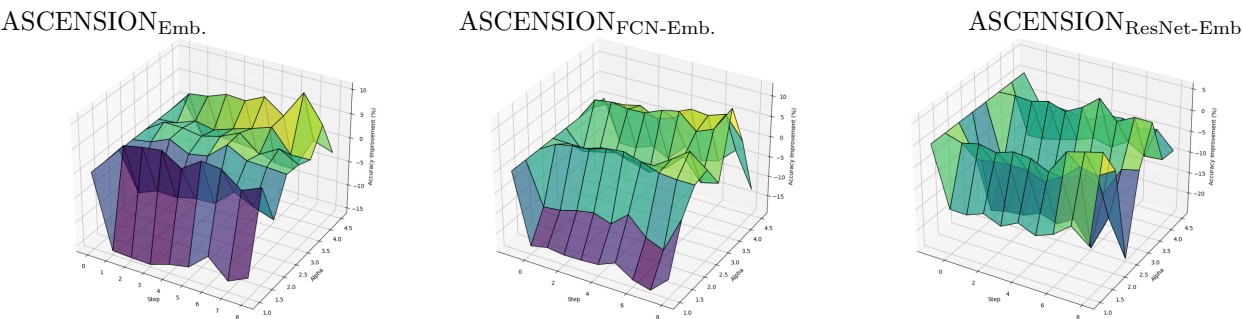

Figure 12: **EOG:** Classifier performance against $\alpha$ and iteration count for **EOGVerticalSignal** datasets.

ASCENSION$_{\text{Emb}}$.         ASCENSION$_{\text{FCN-Emb}}$.         ASCENSION$_{\text{ResNet-Emb}}$.

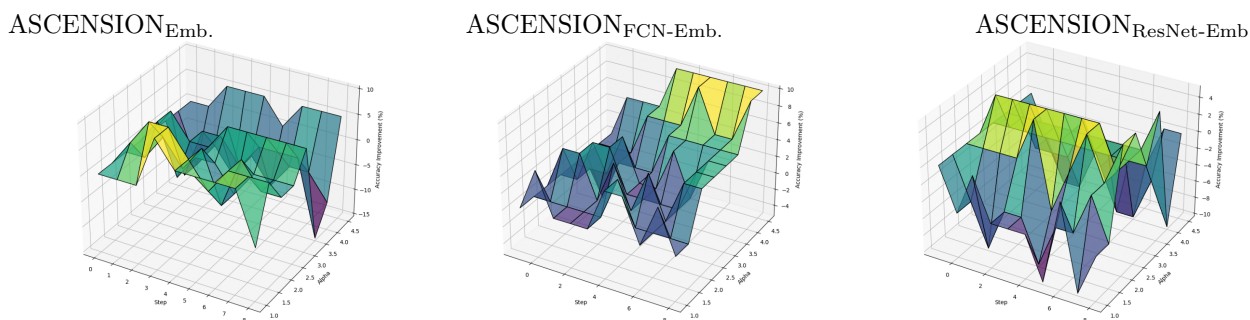

Figure 13: **Image:** Classifier performance against $\alpha$ and iteration count for **BeetleFly** dataset.

Table 23: Accuracy gain ($\overline{Acc}_{\Delta_1 Step}$) after first augmentation step per dataset for the $\alpha$-scaling mechanism and clustering loss ablation studies, along with best pair (iter., $\alpha$) identified per dataset, the train/test ratio (F23), and the train/test discrepancy

| # | Dataset | $\mathcal{L}_{\text{cluster}}$ ablation $\overline{Acc}_{\Delta_1 Step}$ | $\alpha$-scaling ablation (iter., $\alpha$) | Gain | F23 Ratio | Discrepancy $\text{Disp}_{\text{Test}}$ | $\text{Disp}_{\text{Train}}$ |
|---|---|---|---|---|---|---|---|
| D1 | ACSF1 | -0.01 | (1, 2) | -0.01 | 0.96 | $1.01 \times 10^3$ | $1.04 \times 10^3$ |
| D2 | Adiac | -0.01 | (1, 1) | -0.05 | 1.09 | $2.81 \times 10^1$ | $2.25 \times 10^1$ |
| D3 | ArrowHead | -0.05 | (1, 4) | 0.11 | 1.16 | $1.71 \times 10^0$ | $1.88 \times 10^0$ |
| D4 | BME | 0.03 | (1, 2) | -0.00 | 1.13 | $1.64 \times 10^2$ | $1.26 \times 10^2$ |
| D5 | Beef | 0.13 | (1, 3) | 0.27 | 0.94 | $6.40 \times 10^1$ | $6.78 \times 10^1$ |
| D6 | BeetleFly | 0.00 | (1, 1) | 0.32 | 0.89 | $5.79 \times 10^1$ | $5.30 \times 10^1$ |
| D7 | BirdChicken | -0.02 | (1, 1) | 0.00 | 1.30 | $5.28 \times 10^1$ | $5.47 \times 10^1$ |
| D8 | Car | 0.01 | (1, 4) | 0.00 | 1.51 | $3.94 \times 10^2$ | $3.95 \times 10^2$ |
| D9 | CBF | -0.02 | (1, 1) | 0.01 | 1.09 | $6.69 \times 10^4$ | $8.63 \times 10^4$ |
| D10 | ChlorineConcentration | 0.00 | (1, 3) | 0.00 | 1.05 | $9.02 \times 10^1$ | $9.00 \times 10^1$ |
| D11 | CinCECGTorso | -0.00 | (1, 3) | 0.00 | 1.24 | $1.41 \times 10^1$ | $1.40 \times 10^1$ |
| D12 | Coffee | -0.13 | (1, 4) | 0.03 | 1.07 | $8.05 \times 10^0$ | $8.45 \times 10^0$ |
| D13 | Computers | 0.02 | (1, 1) | -0.01 | 0.97 | $1.94 \times 10^2$ | $1.98 \times 10^2$ |
| D14 | Crop | -0.01 | (1, 4) | -0.00 | 1.00 | $1.28 \times 10^4$ | $1.35 \times 10^4$ |
| D15 | DistalPhalanxOutlineAgeGroup | 0.03 | (1, 1) | -0.01 | 1.21 | $6.78 \times 10^1$ | $6.71 \times 10^1$ |
| D16 | DistalPhalanxOutlineCorrect | 0.00 | (1, 2) | -0.00 | 0.96 | $5.31 \times 10^0$ | $6.94 \times 10^0$ |
| D17 | DistalPhalanxTW | -0.02 | (1, 2) | -0.00 | 1.04 | $2.07 \times 10^1$ | $2.07 \times 10^1$ |
| D18 | ECG200 | -0.01 | (1, 2) | 0.01 | 1.21 | $1.28 \times 10^2$ | $1.25 \times 10^2$ |
| D19 | ECG5000 | -0.00 | (1, 2) | 0.02 | 1.19 | $4.17 \times 10^1$ | $4.77 \times 10^1$ |
| D20 | ECGFiveDays | 0.00 | (1, 2) | -0.01 | 1.01 | $5.42 \times 10^1$ | $4.85 \times 10^1$ |
| D21 | EOGHorizontalSignal | 0.00 | (1, 4) | -0.00 | 1.18 | $3.01 \times 10^1$ | $2.65 \times 10^1$ |
| D22 | EOGVerticalSignal | -0.01 | (1, 3) | -0.00 | 0.77 | $6.38 \times 10^3$ | $5.62 \times 10^3$ |
| D23 | Earthquakes | -0.02 | (1, 3) | 0.01 | 1.00 | $1.31 \times 10^2$ | $1.24 \times 10^2$ |
| D24 | ElectricDevices | 0.01 | (1, 2) | -0.01 | 1.10 | $1.02 \times 10^2$ | $9.84 \times 10^1$ |
| D25 | EthanolLevel | -0.00 | (1, 3) | 0.00 | 0.87 | $3.18 \times 10^1$ | $2.10 \times 10^1$ |
| D26 | FaceAll | 0.02 | (1, 1) | 0.01 | 0.97 | $3.58 \times 10^1$ | $3.67 \times 10^1$ |
| D27 | FaceFour | -0.06 | (1, 1) | -0.02 | 1.04 | $5.44 \times 10^1$ | $5.14 \times 10^1$ |
| D28 | FacesUCR | -0.01 | (1, 1) | 0.00 | 1.12 | $1.20 \times 10^3$ | $1.08 \times 10^3$ |
| D29 | Fish | -0.04 | (1, 3) | 0.00 | 1.00 | $1.13 \times 10^3$ | $9.94 \times 10^2$ |
| D30 | FordA | -0.00 | (1, 2) | 0.00 | 0.99 | $3.73 \times 10^3$ | $3.83 \times 10^3$ |
| D31 | FordB | -0.01 | (1, 4) | -0.01 | 0.98 | $2.81 \times 10^0$ | $2.80 \times 10^0$ |
| D32 | FreezerRegularTrain | -0.00 | (1, 3) | 0.00 | 1.01 | $2.47 \times 10^3$ | $3.27 \times 10^3$ |
| D33 | FreezerSmallTrain | 0.03 | (1, 4) | 0.00 | 1.11 | $2.29 \times 10^2$ | $2.35 \times 10^2$ |
| D34 | GunPoint | -0.02 | (1, 2) | 0.00 | 1.04 | $3.83 \times 10^2$ | $3.91 \times 10^2$ |
| D35 | GunPointAgeSpan | -0.00 | (1, 1) | 0.00 | 1.00 | $5.50 \times 10^0$ | $4.90 \times 10^0$ |
| D36 | GunPointMaleVersusFemale | -0.01 | (1, 2) | 0.00 | 1.02 | $3.69 \times 10^1$ | $5.17 \times 10^1$ |
| D37 | GunPointOldVersusYoung | 0.00 | (1, 1) | -0.00 | 1.02 | $5.70 \times 10^2$ | $6.35 \times 10^2$ |
| D38 | Ham | -0.03 | (1, 3) | 0.00 | 1.07 | $4.23 \times 10^2$ | $3.75 \times 10^2$ |
| D39 | HandOutlines | -0.00 | (1, 2) | 0.01 | 0.46 | $1.50 \times 10^2$ | $1.39 \times 10^2$ |
| D40 | Haptics | -0.01 | (1, 4) | 0.01 | 1.08 | $3.27 \times 10^1$ | $2.98 \times 10^1$ |
| D41 | Herring | 0.02 | (1, 2) | -0.01 | 1.01 | $1.02 \times 10^1$ | $1.07 \times 10^1$ |
| D42 | HouseTwenty | 0.00 | (1, 2) | 0.01 | 0.99 | $2.44 \times 10^0$ | $2.79 \times 10^0$ |
| D43 | InlineSkate | -0.02 | (1, 2) | 0.00 | 1.07 | $8.25 \times 10^0$ | $6.80 \times 10^0$ |
| D44 | InsectWingbeatSound | -0.00 | (1, 1) | 0.00 | 1.06 | $7.54 \times 10^2$ | $7.85 \times 10^2$ |
| D45 | InsectEPGRegularTrain | 0.00 | (3, 1) | 0.00 | 1.26 | $1.88 \times 10^1$ | $1.94 \times 10^1$ |
| D46 | InsectEPGSmallTrain | 0.00 | (1, 1) | 0.00 | 1.10 | $1.63 \times 10^2$ | $1.64 \times 10^2$ |
| D47 | ItalyPowerDemand | -0.00 | (1, 3) | 0.00 | 0.95 | $2.75 \times 10^0$ | $2.92 \times 10^0$ |
| D48 | LargeKitchenAppliances | -0.01 | (1, 1) | 0.00 | 1.01 | $3.68 \times 10^1$ | $3.08 \times 10^1$ |
| D49 | Lightning2 | -0.04 | (1, 2) | 0.01 | 1.02 | $5.96 \times 10^1$ | $1.31 \times 10^2$ |
| D50 | Lightning7 | -0.04 | (1, 2) | -0.00 | 0.94 | $3.32 \times 10^1$ | $3.80 \times 10^1$ |
| D51 | Mallat | 0.01 | (1, 4) | 0.00 | 1.11 | $2.40 \times 10^1$ | $2.32 \times 10^1$ |
| D52 | Meat | -0.02 | (1, 3) | -0.02 | 0.94 | $2.80 \times 10^3$ | $1.35 \times 10^3$ |
| D53 | MedicalImages | -0.01 | (1, 2) | -0.01 | 1.00 | $7.57 \times 10^1$ | $8.10 \times 10^1$ |

Table 17: Accuracy gain ($\overline{Acc}_{\Delta_1 Step}$) after first augmentation step per dataset for the $\alpha$-scaling mechanism and clustering loss ablation studies, along with best pair (iter., $\alpha$) identified per dataset, the train/test ratio (F23), and the train/test discrepancy

| # | Dataset | $\mathcal{L}_{\text{cluster}}$ ablation $\overline{Acc}_{\Delta_1 Step}$ | $\alpha$-scaling ablation (iter., $\alpha$) | Gain | F23 Ratio | Discrepancy $\text{Disp}_{\text{Test}}$ | $\text{Disp}_{\text{Train}}$ |
|---|---|---|---|---|---|---|---|
| D54 | MiddlePhalanxOutlineAgeGroup | 0.01 | (1, 3) | -0.01 | 1.12 | $2.70 \times 10^1$ | $2.24 \times 10^1$ |
| D55 | MiddlePhalanxOutlineCorrect | -0.02 | (1, 2) | -0.01 | 0.77 | $1.01 \times 10^6$ | $1.02 \times 10^6$ |
| D56 | MiddlePhalanxTW | -0.03 | (1, 3) | 0.00 | 0.94 | $4.74 \times 10^0$ | $5.02 \times 10^0$ |
| D57 | MixedShapesRegularTrain | 0.00 | (1, 2) | 0.01 | 0.98 | $4.20 \times 10^2$ | $4.76 \times 10^2$ |
| D58 | MixedShapesSmallTrain | 0.01 | (1, 4) | 0.01 | 1.19 | $1.59 \times 10^2$ | $1.55 \times 10^2$ |
| D59 | MoteStrain | 0.00 | (1, 2) | -0.00 | 0.96 | $3.27 \times 10^1$ | $2.60 \times 10^1$ |
| D60 | NonInvasiveFetalECGThorax1 | -0.00 | (1, 1) | -0.00 | 0.98 | $1.31 \times 10^2$ | $1.33 \times 10^2$ |
| D61 | NonInvasiveFetalECGThorax2 | 0.01 | (1, 1) | 0.01 | 0.97 | $2.52 \times 10^0$ | $2.42 \times 10^0$ |
| D62 | OSULeaf | 0.01 | (1, 4) | 0.01 | 0.98 | $8.85 \times 10^3$ | $6.43 \times 10^3$ |
| D63 | OliveOil | 0.00 | (1, 1) | 0.00 | 0.91 | $5.64 \times 10^0$ | $5.94 \times 10^0$ |
| D64 | PhalangesOutlinesCorrect | -0.00 | (1, 2) | -0.00 | 0.95 | $2.02 \times 10^1$ | $2.00 \times 10^1$ |
| D65 | Plane | 0.00 | (1, 1) | 0.00 | 0.94 | $9.58 \times 10^1$ | $1.01 \times 10^2$ |
| D66 | PowerCons | 0.00 | (1, 1) | 0.00 | 1.02 | $2.07 \times 10^4$ | $1.63 \times 10^4$ |
| D67 | ProximalPhalanxOutlineAgeGroup | -0.02 | (1, 1) | -0.02 | 0.94 | $4.70 \times 10^2$ | $7.09 \times 10^2$ |
| D68 | ProximalPhalanxOutlineCorrect | -0.01 | (1, 1) | -0.01 | 0.87 | $1.34 \times 10^1$ | $1.48 \times 10^1$ |
| D69 | ProximalPhalanxTW | -0.00 | (1, 1) | -0.00 | 0.94 | $1.39 \times 10^4$ | $1.36 \times 10^4$ |
| D70 | RefrigerationDevices | -0.01 | (1, 2) | -0.01 | 1.05 | $4.23 \times 10^1$ | $4.01 \times 10^1$ |
| D71 | Rock | -0.01 | (1, 4) | -0.01 | 1.27 | $1.16 \times 10^2$ | $1.11 \times 10^2$ |
| D72 | ScreenType | 0.02 | (1, 2) | 0.02 | 0.88 | $2.18 \times 10^2$ | $2.46 \times 10^2$ |
| D73 | SemgHandGenderCh2 | 0.01 | (1, 3) | 0.01 | 0.97 | $1.47 \times 10^2$ | $1.53 \times 10^2$ |
| D74 | SemgHandMovementCh2 | -0.03 | (1, 1) | -0.03 | 1.00 | $1.23 \times 10^4$ | $1.24 \times 10^4$ |
| D75 | SemgHandSubjectCh2 | -0.01 | (1, 1) | -0.01 | 0.95 | $5.14 \times 10^1$ | $5.28 \times 10^1$ |
| D76 | ShapeletSim | 0.00 | (1, 1) | -0.01 | 1.14 | $1.41 \times 10^4$ | $1.46 \times 10^4$ |
| D77 | ShapesAll | -0.02 | (1, 1) | 0.05 | 0.94 | $4.40 \times 10^1$ | $3.95 \times 10^1$ |
| D78 | SmallKitchenAppliances | 0.01 | (1, 1) | -0.00 | 0.99 | $2.49 \times 10^1$ | $2.61 \times 10^1$ |
| D79 | SmoothSubspace | 0.01 | (1, 2) | -0.00 | 1.04 | $4.86 \times 10^1$ | $3.19 \times 10^1$ |
| D80 | SonyAIBORobotSurface1 | -0.00 | (1, 3) | -0.00 | 1.00 | $8.36 \times 10^0$ | $8.36 \times 10^0$ |
| D81 | SonyAIBORobotSurface2 | -0.01 | (1, 1) | -0.01 | 1.18 | $2.80 \times 10^1$ | $2.36 \times 10^1$ |
| D82 | StarlightCurves | -0.01 | (1, 2) | -0.01 | 1.05 | $5.40 \times 10^4$ | $4.59 \times 10^4$ |
| D83 | Strawberry | -0.01 | (1, 1) | -0.01 | 0.91 | $1.59 \times 10^2$ | $1.56 \times 10^2$ |
| D84 | SwedishLeaf | -0.01 | (1, 1) | -0.01 | 0.96 | $4.69 \times 10^2$ | $4.37 \times 10^2$ |
| D85 | Symbols | -0.02 | (1, 3) | -0.02 | 1.53 | $1.23 \times 10^1$ | $3.72 \times 10^0$ |
| D86 | SyntheticControl | 0.00 | (1, 1) | 0.00 | 1.02 | $2.29 \times 10^2$ | $1.92 \times 10^2$ |
| D87 | ToeSegmentation1 | -0.04 | (1, 1) | -0.04 | 1.18 | $2.19 \times 10^2$ | $2.16 \times 10^2$ |
| D88 | ToeSegmentation2 | 0.02 | (1, 1) | 0.02 | 0.97 | $2.03 \times 10^2$ | $1.72 \times 10^2$ |
| D89 | Trace | -0.01 | (1, 1) | -0.01 | 0.88 | $4.46 \times 10^3$ | $4.41 \times 10^3$ |
| D90 | TwoLeadECG | 0.01 | (1, 3) | 0.01 | 1.00 | $2.28 \times 10^1$ | $2.32 \times 10^1$ |
| D91 | TwoPatterns | -0.01 | (1, 1) | -0.01 | 1.00 | $5.83 \times 10^2$ | $3.90 \times 10^2$ |
| D92 | UMD | 0.02 | (1, 1) | 0.02 | 0.97 | $1.89 \times 10^1$ | $1.89 \times 10^1$ |
| D93 | UWaveGestureLibraryAll | -0.01 | (1, 3) | -0.01 | 1.04 | $5.73 \times 10^1$ | $5.46 \times 10^1$ |
| D94 | UWaveGestureLibraryX | -0.00 | (1, 1) | -0.00 | 1.02 | $6.81 \times 10^1$ | $6.67 \times 10^1$ |
| D95 | UWaveGestureLibraryY | 0.01 | (1, 2) | 0.01 | 1.05 | $2.00 \times 10^1$ | $1.73 \times 10^1$ |
| D96 | UWaveGestureLibraryZ | -0.01 | (1, 1) | -0.01 | 1.06 | $8.64 \times 10^0$ | $9.18 \times 10^0$ |
| D97 | Wafer | -0.00 | (1, 3) | -0.00 | 1.12 | $2.24 \times 10^2$ | $2.30 \times 10^2$ |
| D98 | Wine | 0.01 | (1, 4) | 0.04 | 0.87 | $3.34 \times 10^4$ | $3.33 \times 10^4$ |
| D99 | WordSynonyms | -0.01 | (1, 4) | -0.01 | 1.11 | $4.32 \times 10^3$ | $5.08 \times 10^3$ |
| D100 | Worms | -0.02 | (1, 2) | -0.01 | 0.89 | $1.13 \times 10^2$ | $1.00 \times 10^2$ |
| D101 | WormsTwoClass | -0.03 | (1, 2) | -0.03 | 0.93 | $4.09 \times 10^1$ | $4.26 \times 10^1$ |
| D102 | Yoga | -0.02 | (1, 3) | -0.02 | 1.02 | $3.02 \times 10^4$ | $2.97 \times 10^4$ |

### K.2 Clustering loss ablation study

Table 23 illustrates the step-wise performance evolution given the clustering loss removal via setting the clustering loss weight parameter $\gamma_3$ to zero. To quantify this evolution, we compute the average step-wise improvement using Equation equation 9. Removing the clustering loss results in a performance decrease from 2.1% to $-0.05\%$ with ASCENSION$_{\text{ResNet-Emb}}$, and from 1.2% to $-0.05\%$ with ASCENSION$_{\text{FCN-Emb}}$ (see Table 3).

$$\overline{Acc}_{\Delta_1 Step} = \frac{1}{n-1} \sum_{k=2}^{n} \left( Acc_{Step\_k} - Acc_{Step\_1} \right), \tag{9}$$

where $n$ is the total number of augmentation steps, $Acc_{Step\_1}$ is the accuracy at the first augmentation step (taken as the reference), and the sum averages the per-step accuracy delta over subsequent steps $k = 2, \ldots, n$.

## L  Feature description & Class expansion risk analysis

### L.1  Time series features

We describe hereinafter the 22 TS features (Catch22) presented in (Lubba et al., 2019), and the two additional features (denoted by F23 and F24 below) considered in this study.

**F1: `DN_HistogramMode_5`** Top z-score range based on the highest count from a 5-bin histogram, representing the most frequent distribution range in the dataset.

**F2: `DN_HistogramMode_10`** Similar to DN5, but this considers the top z-score range based on a 10-bin histogram, providing a finer resolution.

**F3: `CO_f1ecac`** Represents the first 1/e crossing of the autocorrelation function, indicating how quickly the autocorrelation of a time series decays.

**F4: `CO_FirstMin_ac`** Identifies the first minimum of the autocorrelation function, which helps analyze the periodicity of the time series.

**F5: `CO_HistogramAMI_even_2_5`** Automutual information for $m = 2$ and $\tau = 5$, capturing the dependency between data points across time.

**F6: `CO_trev_1_num`** This statistic measures time-reversibility, focusing on the differences between successive points in the time series raised to the third power.

**F7: `MD_hrv_classic_pnn40`** Proportion of successive differences in time series values that exceed 0.04 of the standard deviation, indicating rapid fluctuations.

**F8: `SB_BinaryStats_mean_longstretch1`** The longest period where values stay consecutively above the mean, representing persistent trends in the data.

**F9: `SB_TransitionMatrix_3ac_sumdiagcov`** Trace of the covariance of the transition matrix between symbols in a 3-letter alphabet, used to assess transitions in symbolized data.

**F10: `PD_PeriodicityWang_th0_01`** A periodicity measure, indicating how regularly patterns repeat within the time series.

**F11: `CO_Embed2_Dist_tau_d_expfit_meandiff`** Exponential fit to the differences in distances between successive points in a 2-dimensional embedding space, revealing structural relationships.

**F12: `IN_AutoMutlInfoStats_40_gaussian_fmmi`** First minimum of the automutual information function, which gives insight into the periodicity and structure of the time series.

**F13: `FC_LocalSimple_mean1_tauresrat`** Measures the change in correlation length after iteratively differencing the time series, providing insights into the stationarity of the data.

**F14: `DN_OutlierInclude_p_001_mdrmd`** Measures the time intervals between successive extreme events occurring above the mean, indicating patterns of high values.

**F15: `DN_OutlierInclude_n_001_mdrmd`** Similar to DNOp but for extreme events occurring below the mean, highlighting the time intervals between low-value outliers.

**F16: `SP_Summaries_welch_rect_area_5_1`** This computes the total power in the lowest fifth of the frequencies from a Fourier power spectrum, reflecting long-term trends.

**F17: `SB_BinaryStats_diff_longstretch0`** The longest period of successive decreases in the time series, capturing prolonged declining trends.

**F18: `SB_MotifThree_quantile_hh`** Shannon entropy of successive symbol pairs in a 3-letter quantile symbolization, quantifying the complexity of transitions between motifs.

**F19: `SC_FluctAnal_2_rsrangefit_50_1`** Proportion of slower timescale fluctuations that scale with rescaled range fits, indicating long-term memory in the data.

**F20: `SC_FluctAnal_2_dfa_50_1_2_logi_prop`** Proportion of slower timescale fluctuations that scale with detrended fluctuation analysis (DFA) under 50

**F21: `SP_Summaries_welch_rect_centroid`** The centroid of the Fourier power spectrum, which offers a measure of the central frequency or the dominant pattern in the time series.

**F22: `FC_LocalSimple_mean3_stderr`** Calculates the mean error from a rolling 3-sample mean forecast, capturing the volatility of short-term predictions.

**F23: `Train_Test_Ratio`** The ratio of training data to test data in the dataset.

**F24: `Discrepancy_in_Distance`** To estimate the discrepancy in distance between the training and testing set distributions, as defined in Appendix M.

## M  Discrepancy in distance between Training and Test sets

To quantify the discrepancy between training and test distributions, we assess intra-class dispersion using Dynamic Time Warping (DTW) as the distance metric. For each class $k \in \{1, \ldots, K\}$, let the set of time series in class $k$ be denoted by $\mathcal{X}_k = \{x_{k,1}, \ldots, x_{k,n_k}\}$, where $n_k$ is the number of instances in class $k$. Let $\mu_k$ denote the DTW barycenter (i.e., the average sequence under DTW alignment) of $\mathcal{X}_k$. The intra-class dispersion for class $k$ is then defined as $d_k = \frac{1}{n_k} \sum_{i=1}^{n_k} \mathrm{DTW}(x_{k,i}, \mu_k)$

The **Overall dataset dispersion** $D$ is the mean intra-class dispersion across all $K$ classes, defined by $D = \frac{1}{K} \sum_{k=1}^{K} d_k$, and **Discrepancy** is measured as the ratio of test to train set dispersions: $V = \frac{D_{\text{test}}}{D_{\text{train}}}$

The ratio $V$ captures train-test diversity: $V \approx 1$ implies similar variability, $V < 1$ more dispersion in training, and $V > 1$ in testing. Datasets with $V > 1$ are more challenging for generative DA, highlighting the need for augmentation methods that extrapolate beyond the training distribution.

