# OpenReview forum: "ASCENSION: Autoencoder-Based Latent Space Class Expansion for Time Series Data Augmentation"
_TMLR — Under review for TMLR_

### Review · Reviewer_y3Bb · 2026-06-17

**Summary Of Contributions:**

The paper proposes ASCENSION, a VAE-based data augmentation method for time series classification. The motivation is that existing generative DA methods (GANs, diffusion models, VAEs) do not produce consistent gains across diverse domains. ASCENSION's key idea is an α-scaling mechanism that expands the per-class posterior covariances to generate samples beyond the training distribution, combined with a contrastive loss that keeps classes compact and separated so that class identity is preserved during this expansion.

The method is evaluated on 102 UCR datasets with three classifiers (ResNet, FCN, and the Moment foundation model) against eight state-of-the-art DA baselines. The main claims are: an average accuracy gain of roughly 2% (vs. −1.7% for the strongest baseline), non-negative gains on 73.5% of datasets (vs. 50.0% for the second-best method), and being the only method delivering consistent positive gains across all three classifier families without requiring prior knowledge of which method suits a given dataset. An ablation identifies the α-scaling mechanism as the main driver of these gains.

**Audience:**

Yes

**Audience Explanation:**

Yes. The problem of inconsistent DA performance across heterogeneous time series domains is real and relevant, the proposed α-scaling mechanism is a genuinely novel idea, and the theoretical framing (Proposition 1) is of independent interest. The TS classification and DA community would benefit from these findings, provided the empirical claims are brought in line with what the evidence actually supports.

**Broader Impact Concerns:**

None.

**Claims And Evidence:**

No

**Claims Explanation:**

No. While the theoretical contribution (Proposition 1) is sound and the internal ablation is convincing, the central empirical claims are not supported by the evidence as presented.

First, the headline "~2% average gain" does not reflect a consistent effect. From the paper's own per-dataset results (Tables 20–22),  more than half the datasets  fall within a ±1% band, and the top three datasets account for 44% (ResNet) and 63% (FCN) of the total gain (Beef, BirdChicken, EthanolLevel). Two of these three are also very small: [Beef](https://www.timeseriesclassification.com/description.php?Dataset=Beef) has 30 train and 30 test samples across 5 classes (6 per class), and [BirdChicken](https://www.timeseriesclassification.com/description.php?Dataset=BirdChicken) has 20 train and 20 test samples across 2 classes (10 per class).So on two of the three classifiers the mean is therefore driven by a handful of (partly very small) datasets rather than being representative of a typical dataset. On Moment classifier the gains are somewhat more frequent (median +1.0%) but still small, with the majority of datasets within ±2%.

Second, the consistency claim ("the only method with consistent gains across all three classifier families") is only partially borne out by the authors' own significance tests (App. F): ASCENSION is statistically indistinguishable from KoVAE on ResNet and shares a rank band with four baselines on FCN; it is clearly ahead only on Moment, which is also the classifier whose evaluation protocol is left unspecified. This comparison is moreover not on equal footing: ASCENSION is reported at its per-dataset best configuration selected on the test set (Tables 20–22), while the baselines use fixed author-recommended settings, so the gap is overstated in ASCENSION's favour. Under a single fixed configuration (Table 7) ASCENSION's gain already drops to +1.7/+1.0/+0.6%.

Finally, no variance over training seeds is reported anywhere, so it cannot be established that the small per-dataset gains exceed training noise. Details are in Requested Changes.

**Requested Changes:**

Below, [Critical] marks changes I consider necessary to support the paper's claims and [Would strengthen] marks changes that would improve the work without being decisive.

 **[Critical] 1. Make the primary comparison fair, and recompute the supporting analyses on it.** In Table 1, ASCENSION is reported at its per-dataset best configuration selected on the test set, while the baselines use fixed author-recommended settings. This selects hyperparameters on test data and is not a like-for-like comparison. It inflates ASCENSION's apparent advantage. The single fixed configuration (currently Table 7, +1.7/+1.0/+0.6%) should become the main result in the body, and all analyses that currently build on the test-selected results (including the ablations and the significance tests) should be recomputed under that same configuration, so that every reported conclusion rests on the fair setting. The remaining points below should be read against this fair configuration.

**[Critical] 2. Report the distribution of per-dataset gains, not only the mean, and adjust the claims accordingly.** Even under the optimistic test-selected configuration, the "~2% average gain" is not representative: from Tables 20–22 the median gain is already 0.0% on both ResNet and FCN, 53/102 datasets fall within a ±1% band, and the top three datasets contribute 44% (ResNet) and 63% (FCN) of the total (two of them, Beef and BirdChicken, very small). The abstract's statement of "consistent gains on a majority of the datasets" is contradicted by these same tables: with a median of 0% on ResNet and FCN, the majority of datasets show no gain rather than a gain. (The separate "non-negative gains on 73.5%" figure is not affected by this, since zero counts as non-negative.) On Moment classifier the gains are more evenly spread but remain small (61/102 within ±2%). Under the fair configuration of point 1 this picture can only become weaker and should be shown explicitly: please report medians and a concentration measure (e.g. share of total gain from the top-k datasets), and reformulate the headline claims so they reflect that on two of three classifiers the effect is carried by a few datasets rather than being typical.

**[Critical] 3. Align the across-classifier consistency claim with the significance tests, and move them into the body.** The authors' own Friedman/Nemenyi analysis (App. F) shows ASCENSION statistically indistinguishable from KoVAE on ResNet and sharing a rank band with four baselines on FCN, with clear separation only on Moment. The claim of being "the only method with consistent gains across all three classifier families" overstates this, and the tests are themselves computed on the test-selected configuration, so they already overstate ASCENSION's standing. Please recompute them on the fair configuration, bring the wording in line with the result, and move these tests into the main text, since they are central to the headline claim.

**[Critical] 4. Report variability over training seeds.** No variance is reported anywhere. Given that the median gain is 0% and most per-dataset gains lie within ±1%, it cannot be established that the reported improvements exceed training noise; this concern is independent of the configuration issue and compounds it. Please run multiple seeds and report standard deviations or confidence intervals for the headline numbers and, ideally, per dataset.

**[Critical] 5. Specify how Moment is used.** Moment is the only classifier on which the superiority claim holds, yet the paper does not describe how it is used as a classifier. Since a pretrained model requires some form of per-dataset adaptation to produce class predictions, please specify exactly what was done and how the augmented samples enter that procedure. Without it, the strongest piece of evidence for the main claim is not interpretable.

**[Critical] 6. Specify the VAE architecture.** The encoder/decoder layer type (recurrent, convolutional, transformer) and the latent dimension are not given anywhere in the paper. This leaves the method underspecified, and it also makes it impossible to tell whether the cVAE comparison (App. D) use the same architecture as ASCENSION. Please report the architecture and state whether the App. D variants share it.

**[Critical] 7. Clarify the remaining evaluation details.** Please state explicitly whether best-epoch-on-test selection is applied to the baselines as well as to ASCENSION.

**[Would strengthen] 8. Relate gains to per-class sample size.** About half the retained datasets have fewer than 50 samples per class (32 have fewer than 20), which is little for training a generative model. Plotting the gain against samples-per-class would test whether the few datasets driving the mean are precisely the very small ones, and would clarify where the method is and is not applicable.

**[Would strengthen] 9. Motivate the Moment and embedded-classifier configurations.** The embedded variants (Section 4.1.3) and the choice of Moment as a third classifier are introduced without explaining what they add over the default setup. A brief motivation would help the reader interpret their role, particularly since the embedded variants carry the main ablation.

---

### Review · Reviewer_Hzd4 · 2026-07-01

**Summary Of Contributions:**

Summary:

The authors study data augmentation for time series classification models.

They propose ASCENSION, a method that trains a VAE and then scales the variance of the class clusters to expand the latent distribution into under-represented neighborhoods.

They evaluate their method on 102 univariate UCR datasets and compare with multiple baselines. Data augmentation using their method improves the performance for most datasets, and improves the average performance across all 102.






***
Strengths:
- The paper is well written overall.
- The proposed method is quite interesting and makes sense overall, and could potentially be applicable also to other domains than time series.
- The results in Section 4.1 (Table 1) seem solid, with consistently good performance of the proposed method compared to multiple baselines.





***
Weaknesses:
- I don't think it's clearly stated how e.g. the value of alpha and the number of iterations are selected for each individual dataset in the results. _"We sweep [...] and augmentation iterations {1, . . . , 9}"_ is mentioned in Section 4.2, but I don't see any actual results for this.
- The analysis in Section 4.3 is somewhat unclear, I'm not quite sure how to interpret Figure 5 and these results, or exactly what conclusions can be drawn from this.









***
Questions/suggestions:
- Not sure that Section 2.2 needs to be this long, not quite sure how much it actually adds. I think it could be more useful to have a shorter and more focused related work here, and then move the full version to the appendix.
- It's not clear to me how many new examples are generated in each iteration in Algorithm 1? And specifically when doing just a single iteration, how many examples are generated then? Do you double the size of the dataset?
- Overall, I think the iterative approach of the proposed method should be analyzed/discussed more. Why do you need to use more than a single iteration (intuitively, why does it make sense that this should help?)? How much does this actually improve the results? Does this improve the results consistently across different datasets? If you have a fixed budget in terms of number of generated examples (100, for example), is it better to e.g. generate 25 samples per iteration over 4 iterations, than to generate all 100 in a single iteration?
- Does the performance of your method perhaps depend on the number of classes? Might it be less effective for large number of classes?
- Could the proposed method be extended also to regression problems?
- The proposed method is general and could in principle be applied to other domains, not just time series data, right? Have you explored applying this e.g. to image-based datasets?











***
Minor things:
- Very minor thing, but there is quite a lot of white-space between figure and caption for Fig 6, between rows as well. Figure 5 could also be made a bit tighter.
- The last 3 sentences of the first paragraph in Introduction is basically repeated in Section 2.1.

**Audience:**

Yes

**Audience Explanation:**

I think the proposed method is quite interesting and that it potentially could be quite broadly applicable/useful.

**Broader Impact Concerns:**

No concerns.

**Claims And Evidence:**

No

**Claims Explanation:**

I think the paper just requires a few clarifications before publication.

**Requested Changes:**

This is a quite interesting and well-written paper that I definitely think could be relevant for the TMLR audience.

However, I think the current version would benefit from some clarifications and modifications, see "Weaknesses" and "Questions/suggestions" above.

---

### Comment · Action_Editor_jEff · 2026-07-20
**AE comment after reviews**

Dear authors and reviewers,

Thank you to the reviewers for the thoughtful and detailed evaluations, and to the authors for their patience.

At this stage, we have received two reviews, and I encourage the discussion to proceed based on the feedback currently available.

Authors, please respond to the reviewers' comments in the following two weeks, with particular attention to the points that appear most central to the evaluation, including:
 - the fairness of the experimental protocol and the strength of the empirical claims (in particular, model selection, statistical significance, and the interpretation of the reported performance gains),
 - clarification of methodological and implementation details (e.g., hyperparameter selection, VAE architecture, the use of the Moment classifier, and other aspects of the evaluation protocol),
 - the additional analyses and clarifications requested regarding the iterative augmentation process, parameter choices, and the conditions under which the proposed method is expected to perform well.

Reviewers, after the authors have responded, please continue the discussion and indicate whether the responses address your concerns or whether any issues remain that are important for the final evaluation.

Thank you all for your constructive participation in the review process.